# NOISE-ADAPTIVE DIFFUSION SAMPLING FOR INVERSE PROBLEMS WITHOUT TASK-SPECIFIC TUNING

**Yingzhi Xia**[1*]  **Setthakorn Tanomkiattikun**[1,3*]  **Liangli Zhen**[1†]  **Zaiwang Gu**[2]

[1]Institute of High Performance Computing, Agency for Science, Technology and Research, Singapore
[2]Institute for Infocomm Research, Agency for Science, Technology and Research, Singapore
[3]Johns Hopkins University
{Xia_Yingzhi, zhen_liangli, Gu_Zaiwang}@a-star.edu.sg   stanomk1@jhu.edu

## ABSTRACT

Diffusion models (DMs) have recently shown remarkable performance on inverse problems (IPs). Optimization-based methods can fast solve IPs using DMs as powerful regularizers, but they are susceptible to local minima and noise overfitting. Although DMs can provide strong priors for Bayesian approaches, enforcing measurement consistency during the denoising process leads to manifold infeasibility issues. We propose Noise-space Hamiltonian Monte Carlo (N-HMC), a posterior sampling method that treats reverse diffusion as a deterministic mapping from initial noise to clean images. N-HMC enables comprehensive exploration of the solution space, avoiding local optima. By moving inference entirely into the initial-noise space, N-HMC keeps proposals on the learned data manifold. We provide a comprehensive theoretical analysis of our approach and extend the framework to a noise-adaptive variant (NA-NHMC) that effectively handles IPs with unknown noise type and level. Extensive experiments across four linear and three nonlinear inverse problems demonstrate that NA-NHMC achieves superior reconstruction quality with robust performance across different hyperparameters and initializations, significantly outperforming recent state-of-the-art methods. The code is available at https://github.com/NA-HMC/NA-HMC.

## 1 INTRODUCTION

Inverse problems (IPs) have wide applications in many domains, including computer vision (Janai et al., 2021; Quan et al., 2024), protein science (Yi et al., 2023; Ouyang-Zhang et al., 2023; Yang et al., 2019), medical imaging (Song et al., 2022b; Chu et al., 2025; Dao et al., 2024), scientific computing (Zheng et al., 2025; Xia & Zabaras, 2022; Xu et al., 2024). The goal is to reconstruct an unknown $x \in \mathbb{R}^n$ from noisy measurements $y \in \mathbb{R}^m$:

$$y = \mathcal{A}(x) + \eta, \tag{1}$$

where $\mathcal{A}$ is a known forward operator, and $\eta \in \mathbb{R}^m$ is additive noise. Diffusion models (DMs) have recently shown powerful capabilities in modeling complex data distributions, which can provide a powerful class of priors for high-dimensional data $x$ in solving IPs.

Existing diffusion-based methods have demonstrated remarkable success across diverse inverse problems (Chung et al., 2023; Daras et al., 2024; Zheng et al., 2025; Song et al., 2022b). Although remarkable progress has been made, as illustrated in Figure 1, current diffusion-based methods suffer from three complementary limitations and issues: (1) Iterative guidance methods such as DPS (Chung et al., 2023), DDRM (Kawar et al., 2022), DDNM(Wang et al., 2023), ΠGDM (Song et al., 2023), and TMPD (Boys et al., 2024) use the likelihood term to shift intermediate images directly, systematically pushing intermediate states off the learned data manifold and violating the training-time noise-conditioning of the denoiser, resulting in various failure reconstructions like accumulated artifacts as shown in Figure 1 (a). (2) Stochastic MAP methods that optimize in image space, including ReSample (Song et al., 2024), DiffPIR (Zhu et al., 2023), DAPS (Zhang et al., 2024), SITCOM

---

*Equal contribution.
† Corresponding author.

(Alkhouri et al., 2025a), and DIIP (Chihaoui & Favaro, 2025a) can match $y$ well with very sharp details but require carefully tuned hyperparameters to not overfit to noise. This limits their effectiveness in high or unknown noise settings. (3) Deterministic MAP methods that optimize in the DM noise space (DMPlug, (Wang et al., 2024)) remove randomness but often get stuck in a single mode, especially in severely ill-posed problems like phase retrieval, due to a lack of posterior exploration. In short, enforcing data consistency mid-diffusion can break prior adherence, while optimizing only for fidelity leads to overfitting or mode collapse.

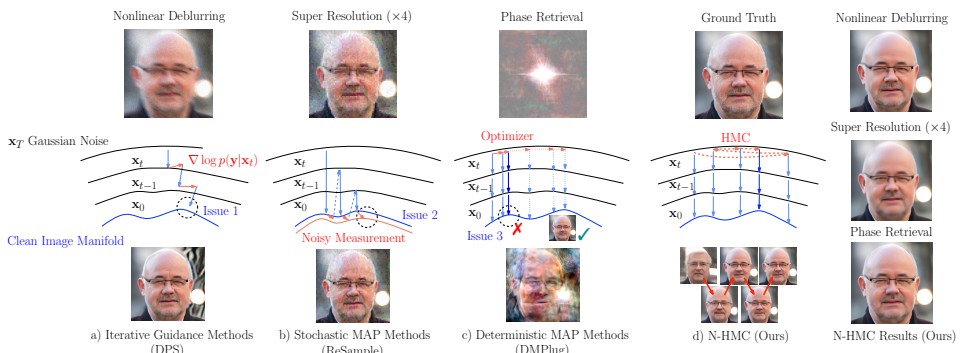

Figure 1: Comparison of existing methods and their limitations with the N-HMC method. (a) Iterative Guidance Methods (*DPS*) lead to *manifold infeasibility*. (b) Stochastic MAP methods (*ReSample*) (Song et al., 2024) are susceptible to *overfitting to noise*. (c) Deterministic MAP methods (*DMPlug*) (Wang et al., 2024) become *trapped in a local mode*. (d) Our method performs sampling in the noise space $x_T$ and maps samples to images via a deterministic mapping $\mathbf{x}_0 = \mathcal{D}(\mathbf{x}_T)$.

Sampling from the full posterior ensures that the learned prior acts automatically as a regularizer, while an annealing schedule for noise standard deviation $\sigma_y$ promotes efficient exploration and prevents the sampler from being trapped in early local modes. Importantly, the method relies only on a fixed set of hyperparameters that remain constant across tasks, datasets, levels of measurement noise, avoiding the repeated tuning required by many existing approaches.

To address the practical challenges that the measurement noise level is often unknown, we further introduce a Noise-Adaptive N-HMC (NA-NHMC). Instead of requiring a fixed noise level, we take a principled Bayesian approach, placing a non-informative prior on the noise variance and marginalizing it out. This yields a parameter-free likelihood term that automatically adapts to the true underlying noise in the measurements. As shown in experiments, this allows NA-NHMC to achieve robust, high-quality reconstructions across varying and even unknown noise types and levels without any task-specific hyperparameter tuning. In contrast, the performance of other methods depends on the hyperparameters listed in Section A.5, which were specifically tuned for Gaussian noise. Our key *contributions* include: (1) In Section 3.1, we propose N-HMC, a posterior sampling method that addresses the three key limitations of existing state-of-the-art (SOTA) approaches. We further analyze its sampling behavior and provide a theoretical guarantee of its robustness to measurement noise in Section 3.2 and Appendix A.2. (2) In Section 3.3, we extend our method to settings with unknown noise types and levels. We show that it outperforms SOTA methods on most metrics (Section 4.3), especially for non-linear and high noise problems. (3) In extensive experiments, NA-NHMC method solves diverse inverse problem tasks under unknown noise types and levels without any task- or noise-specific hyperparameter tuning, in contrast to many existing methods. (4) We demonstrate in Section 4.1 that the annealing schedule for $\sigma_y$ helps promote early exploration and prevent local-mode collapse, especially in severely ill-posed tasks like phase retrieval.

## 2 PRELIMINARIES

### 2.1 DIFFUSION MODELS FOR INVERSE PROBLEMS

Daras et al. (2024) broadly classifies methods for solving inverse problems (IPs) into two categories. The first is maximum a posteriori (MAP) inference, which aims to find the single most probable $x$. An alternative is the Bayesian framework, where the goal becomes generating plausible reconstruc-

tions by sampling from the posterior distribution $p(\boldsymbol{x}|\boldsymbol{y})$, where $p(\boldsymbol{x}|\boldsymbol{y})$ can be decomposed into the prior $p(\boldsymbol{x})$ and the likelihood $p(\boldsymbol{y}|\boldsymbol{x})$. MAP delivers fast optimization, but struggles with high noise and multimodal posteriors, easily converging to local minima. In contrast, the Bayesian approach samples from $p(\boldsymbol{x}|\boldsymbol{y})$ to generate plausible reconstructions, quantify uncertainty, and handle multimodality. Both approaches critically depend on powerful prior models like DMs that encode the complex statistical structure of complex data and prior knowledge.

Most diffusion-based approaches to IPs are based on the denoising diffusion probabilistic models (DDPM) framework (Ho et al., 2020; Song & Ermon, 2020). The framework consists of forward and reverse diffusion processes. The forward process gradually corrupts the clean images $\boldsymbol{x}_0$ towards standard Gaussian noise $\boldsymbol{x}_T$. This process can be described by a stochastic differential equation (SDE), $d\boldsymbol{x} = -\frac{\beta_t}{2}\boldsymbol{x}dt + \sqrt{\beta_t}d\boldsymbol{w}$, where $\boldsymbol{w}$ is the standard Wiener process. In practice, the process is discretized via a variance schedule $\{\beta_t\}_{t=1}^T$, forming a Markov chain:

$$q(\boldsymbol{x}_{1:T}|\boldsymbol{x}_0) \coloneqq \prod_{t=1}^T q(\boldsymbol{x}_t|\boldsymbol{x}_{t-1}), \qquad q(\boldsymbol{x}_t|\boldsymbol{x}_{t-1}) \coloneqq \mathcal{N}(\boldsymbol{x}_t; \sqrt{1-\beta_t}\boldsymbol{x}_{t-1}, \beta_t\boldsymbol{I}) \qquad (2)$$

In order to generate clean images, the reverse process begins with a noisy sample $\boldsymbol{x}_T \sim \mathcal{N}(\boldsymbol{x}_T; \boldsymbol{0}, \boldsymbol{I})$, and recursively refines it according to the reverse SDE, $d\boldsymbol{x} = -\frac{\beta_t}{2}\boldsymbol{x}dt - \beta_t\nabla_{\boldsymbol{x}}\log p_t(\boldsymbol{x})dt + \sqrt{\beta_t}d\overline{\boldsymbol{w}}$, where $\overline{\boldsymbol{w}}$ is the time-reversed standard Wiener process, and $p_t(\boldsymbol{x})$ is the marginal probability of the noisy manifold at time $t$. $\nabla_{\boldsymbol{x}}\log p_t(\boldsymbol{x})$ is called the score function and is usually approximated by a neural network $\theta$ trained through score-matching methods.

Using the same discretization, clean images can be generated from the prior using an iterative denoising process.

$$p_\theta(\boldsymbol{x}_{0:T}) \coloneqq p(\boldsymbol{x}_T) \prod_{t=1}^T p_\theta(\boldsymbol{x}_{t-1}|\boldsymbol{x}_t), \qquad p_\theta(\boldsymbol{x}_{t-1}|\boldsymbol{x}_t) \coloneqq \mathcal{N}(\boldsymbol{x}_{t-1}; \boldsymbol{\mu}_\theta(\boldsymbol{x}_t, t), \boldsymbol{\Sigma}_\theta(\boldsymbol{x}_t, t)) \qquad (3)$$

Building on the DDPM framework, to accelerate the denoising process, Song et al. (2022a) proposes Denoising Diffusion Implicit Models (DDIM), which define a non-Markovian and fully deterministic forward/reverse process ($\beta_t = 0$). Unlike DDPM, which injects stochasticity at each step to improve robustness, DDIM iteratively maps the initial noise $\boldsymbol{x}_T$ to a clean sample $\boldsymbol{x}_0$ via a deterministic trajectory. For our method, this property is particularly beneficial, as it allows us to consider the entire reverse process as a deterministic mapping from $\boldsymbol{x}_T$ to $\boldsymbol{x}_0$.

Among successful DM-based methods for inverse problems, DPS and its variants (Chung et al., 2023; Kawar et al., 2022; Wang et al., 2023; Song et al., 2023; Chung et al., 2022) are best-known reconstruction algorithms. But they suffer from approximation errors from Tweedie's formula corrections. To mitigate noise sensitivity, TMPD incorporates second-order information to correct the guidance trajectory; however, like other iterative methods, it relies on modifying intermediate states, which risks drifting off the learned manifold. SITCOM (Alkhouri et al., 2025b) operates on the noisy image at each diffusion step and enforces a triple-consistency constraint: data fidelity, backward consistency with the diffusion posterior mean, and forward consistency along the diffusion trajectory. DIIP (Chihaoui & Favaro, 2025b) updates the initial noise $\boldsymbol{x}_T$ with data fidelity gradients after the standard diffusion sampling process. DMPlug (Wang et al., 2024) proposes a noise-space formulation but treats inverse problems as optimization tasks, making it *sensitive to noise*. While early stopping can mitigate this, its criterion is task- and noise-dependent. At the high noise levels considered here, the optimizer often becomes *trapped in a local mode*, rendering early stopping ineffective. Similar behavior is observed in other Maximum a Posteriori (MAP) methods such as ReSample (Song et al., 2024), which optimizes directly in clean-image space (leading to noisy or blurry images under early stopping). DAPS (Zhang et al., 2024), despite being formulated as a posterior sampling method, uses a heuristic $\hat{\sigma}_y$ that is much smaller than its true value to strengthen the consistency of the measurement. This deviation from true posterior sampling makes DAPS effectively MAP-like, inheriting the same sensitivity to noise.

## 2.2 HAMILTONIAN MONTE CARLO (HMC)

Hamiltonian Monte Carlo (HMC) (Duane et al., 1987) is an MCMC (Metropolis et al., 1953) sampling method that utilizes a fictitious momentum variable and simulates Hamiltonian dynamics to

efficiently explore distant regions. Due to its superior scaling properties in high dimensions compared to other simpler Metropolis methods Brooks et al. (2011), HMC is particularly well suited for sampling in high-dimensional space, such as the $3 \times 256 \times 256$ pixel space of images.

The Hamiltonian is defined as $H = U + V$, where $U = -\log p(\boldsymbol{x})$ and $V = \frac{1}{2}\boldsymbol{v}^\top \boldsymbol{M}^{-1}\boldsymbol{v}$. Then, we discretize the trajectory using the leapfrog integrator. For a single leapfrog step with step size $\delta$, we have

$$\boldsymbol{v}(t + \delta/2) = \boldsymbol{v}(t) - \frac{\delta}{2}\frac{\partial U}{\partial \boldsymbol{x}}\bigg|_{\boldsymbol{x}(t)}, \tag{4}$$

$$\boldsymbol{x}(t + \delta) = \boldsymbol{x}(t) + \delta \boldsymbol{M}^{-1}\boldsymbol{v}(t + \delta/2), \tag{5}$$

$$\boldsymbol{v}(t + \delta) = \boldsymbol{v}(t + \delta/2) - \frac{\delta}{2}\frac{\partial U}{\partial \boldsymbol{x}}\bigg|_{\boldsymbol{x}(t+\delta)}, \tag{6}$$

where $\boldsymbol{v}(0) \sim \mathcal{N}(\boldsymbol{v}; \boldsymbol{0}, \boldsymbol{M})$. This process is repeated $L$ times to form a full trajectory. Due to a discretization error, the Hamiltonian is no longer preserved, which introduces bias and violates the detailed balance. To correct for this, a Metropolis-Hastings (MH) correction step is applied at the end of each trajectory with acceptance probability of $\alpha = \min(1, \exp(-H_1 + H_0))$, where $H_0, H_1$ denotes the initial and proposed Hamiltonian, respectively.

## 3 METHODOLOGY

In this section, we propose a posterior sampling method, Noise-space Hamiltonian Monte Carlo (N-HMC), to solve IPs with pretrained DMs. We show its derivation in Section 3.1 and discuss its robustness to measurement noise in Section 3.2. In Section 3.3, our method is modified to allow for unknown types and levels of measurement noise.

### 3.1 NOISE-SPACE HAMILTONIAN MONTE CARLO (N-HMC)

The goal in solving inverse problems is to sample from the posterior distribution $p(\boldsymbol{x}_0|\boldsymbol{y}) \propto p(\boldsymbol{x}_0)p(\boldsymbol{y}|\boldsymbol{x}_0)$. Since direct sampling from $p(\boldsymbol{x}_0)$ is intractable, pretrained diffusion models are employed to provide a powerful prior. Standard diffusion-based approaches draw $\boldsymbol{x}_T \sim p(\boldsymbol{x}_T)$ from a Gaussian noise prior and iteratively denoise through intermediate timesteps, aiming to sample from $p(\boldsymbol{x}_T \mid \boldsymbol{y}), p(\boldsymbol{x}_{T-1} \mid \boldsymbol{y}), \ldots, p(\boldsymbol{x}_0 \mid \boldsymbol{y})$ in sequence. The key challenge is evaluating the intractable likelihood $p(\boldsymbol{y}|\boldsymbol{x}_t)$ at each intermediate timestep $t$. To address this, iterative guidance methods (Kawar et al., 2022; Wang et al., 2023; Chung et al., 2023; Song et al., 2023; Rozet et al., 2024; Song et al., 2024; Zhang et al., 2024) introduce approximations and apply likelihood corrections of $\boldsymbol{x}_t \leftarrow \boldsymbol{x}_t + \eta \nabla_{\boldsymbol{x}_t} \log p(\boldsymbol{y}|\boldsymbol{x}_t)$. However, these gradient-based corrections systematically push intermediate states $\boldsymbol{x}_t$ away from the distribution on which the denoiser is trained, leading to what we refer to as the *manifold feasibility problem*. Following SITCOM (Alkhouri et al., 2025b), we formalize this issue as:

**Definition 3.1** (Manifold Feasibility). *For a pretrained diffusion model, let $p_t(\boldsymbol{x}_t)$ denote the marginal distribution at noise level $t$, and let $\mathcal{M}_t$ be its high-probability generative manifold. An inverse-problem solver maintains* manifold feasibility *if the intermediate states $\{\boldsymbol{x}_t\}$ fed into the denoiser remain close to $\mathcal{M}_t$ for all $t$, ensuring the final reconstruction $\boldsymbol{x}_0$ lies on the learned data manifold.*

Geometrically, standard guidance methods update $\mathbf{x}_t$ using the likelihood gradient $\nabla_{\mathbf{x}_t} \log p(\mathbf{y}|\mathbf{x}_t)$. In high-dimensional spaces, this gradient vector often contains components orthogonal to the local tangent space of the data manifold $\mathcal{M}_t$. Consequently, adding this gradient systematically pushes the state $\mathbf{x}_t$ into low-probability regions (off-manifold), feeding out-of-distribution inputs to the denoiser and causing accumulated artifacts as shown in Figure 1 (a). To avoid such approximations, we propose posterior sampling by drawing from the initial noise space. The sampled noise is then unconditionally denoised to a clean image. We adopt unconditional DDIM for the denoising process, which treats the entire denoising trajectory as a deterministic mapping $\hat{\mathbf{x}}_0 = \mathcal{D}(\mathbf{x}_T)$, so the problem becomes evaluating the posterior distribution of noise (Xia et al., 2023), i.e., $p(\boldsymbol{x}_T|\boldsymbol{y})$. We refer to our approach as *noise-space sampling* because HMC updates are performed exclusively on the initial noise $x_T \sim \mathcal{N}(0, I)$. This differs from image-space and iterative guidance methods that

directly modify intermediate states $\boldsymbol{x}_t$ using measurement-consistency gradients. Sampling from the noise space offers two advantages: (i) the prior $p(\boldsymbol{x}_T)$ is a simple Gaussian distribution, and (ii) the likelihood $p(\boldsymbol{y}|\boldsymbol{x}_T) = p(\boldsymbol{y}|\mathcal{D}(\boldsymbol{x}_T))$ is directly accessible without intermediate approximations.

We use HMC for efficient posterior sampling in the noise space. To strictly justify our sampling objective, we formulate the inference process as a latent variable model where the initial noise $x_T$ is the sole latent variable. We treat the unconditional DDIM process with $N$ steps as a deterministic parameterized generator function, denoted as $\mathcal{D} : \mathbb{R}^n \to \mathbb{R}^n$, which maps $\boldsymbol{x}_T$ to a clean image $\hat{\boldsymbol{x}}_0 = \mathcal{D}(\boldsymbol{x}_T)$. Under this formulation, the measurement generation process is defined by $\boldsymbol{x}_T \xrightarrow{\mathcal{D}} \hat{\boldsymbol{x}}_0 \xrightarrow{\mathcal{A},\eta} \boldsymbol{y}$. Consequently, the conditional distribution of $y$ given $x_T$ depends entirely on the generated image $\hat{\boldsymbol{x}}_0$. The likelihood term is thus mathematically exact: $p(\boldsymbol{y}|\boldsymbol{x}_T) = p(\boldsymbol{y}|\hat{\boldsymbol{x}}_0 = \mathcal{D}(\boldsymbol{x}_T)) = \mathcal{N}(\boldsymbol{y}; \mathcal{A}(\mathcal{D}(\boldsymbol{x}_T)), \sigma_y^2 I)$. This allows us to perform posterior sampling directly in the noise space using the exact gradient of the likelihood with respect to $x_T$. Then we can compute the conditional score using Bayes' rule:

$$\nabla_{\boldsymbol{x}_T} \log p(\boldsymbol{x}_T|\boldsymbol{y}) = \nabla_{\boldsymbol{x}_T} \log p(\boldsymbol{x}_T) + \nabla_{\boldsymbol{x}_T} \log p(\boldsymbol{y}|\boldsymbol{x}_T). \tag{7}$$

Since $\boldsymbol{x}_T$ is Gaussian noise in the DDIM framework, the first term is simply

$$\nabla_{\boldsymbol{x}_T} \log p(\boldsymbol{x}_T) = -\nabla_{\boldsymbol{x}_T} \frac{\|\boldsymbol{x}_T\|^2}{2} = -\boldsymbol{x}_T. \tag{8}$$

For the case of Gaussian measurement noise, if the noise level $\sigma_y^2$ is known, the likelihood term becomes

$$\nabla_{\boldsymbol{x}_T} \log p(\boldsymbol{y}|\boldsymbol{x}_T) = \nabla_{\boldsymbol{x}_T} \log p(\boldsymbol{y}|\mathcal{D}(\boldsymbol{x}_T)) = -\nabla_{\boldsymbol{x}_T} \frac{\|\boldsymbol{y} - \mathcal{A}(\mathcal{D}(\boldsymbol{x}_T))\|^2}{2\sigma_y^2}. \tag{9}$$

We define $p(\boldsymbol{y}|\boldsymbol{x}_T) = p(\boldsymbol{y}|\mathcal{D}(\boldsymbol{x}_T))$ by viewing the denoising trajectory as a deterministic mapping $\hat{\mathbf{x}}_0 = \mathcal{D}(\mathbf{x}_T)$. This term can be computed directly using automatic differentiation. Because $\mathcal{D}(\boldsymbol{x}_T)$ results from a multi-step denoising process, backpropagating through multiple score networks can be computationally expensive. Following Wang et al. (2024), we illustrate in Appendix A.9 that accurate samples can still be obtained with as few as two denoising steps.

Once the conditional score $\nabla_{\boldsymbol{x}_T} \log p(\boldsymbol{x}_T|\boldsymbol{y})$ is computed, our method proceeds with standard Hamiltonian Monte Carlo (HMC) sampling. We use the identity matrix as the mass matrix for momentum sampling. During implementation, we observed that the initial noise may lie in regions of very low posterior probability, which forces HMC to adopt a tiny step size in order to maintain a proper acceptance rate. To address this issue, we use an annealing schedule for $\sigma_y$, allowing $\boldsymbol{x}_T$ to explore the noise space more freely with a larger step size in the start-up stage. Once $\sigma_y$ gradually declines to the target level, posterior samples are collected. The complete procedure is summarized in Algorithm 1, along with the unconditional DDIM denoising process in Algorithm 2.

## 3.2 ROBUSTNESS TO MEASUREMENT NOISE

An additional benefit of N-HMC over MAP methods is that the Gaussian prior acts as a regularization term in the noise space, keeping the noise vector $\boldsymbol{x}_T$ close to the hypersphere of radius $\sqrt{n}$. Therefore, N-HMC produces samples that are *robust to measurement noise*, as justified by Proposition 1. For simplicity, we assume Gaussian measurement noise and that the forward operator $\mathcal{A}$ is approximately linear along the clean image manifold.

**Proposition 1.** Assume that the distribution of the decoded sample $\boldsymbol{x}_0$ around the ground truth $\boldsymbol{x}_0^*$ is well-approximated by a Gaussian distribution $p_\theta(\hat{\boldsymbol{x}}_0) \approx \mathcal{N}(\hat{\boldsymbol{x}}_0; \boldsymbol{x}_0^*, \sigma_0^2 \boldsymbol{I}_n)$. Then, the residual $\boldsymbol{y} - \boldsymbol{A}\hat{\boldsymbol{x}}_0$ satisfies

$$\mathbb{E}_{(\hat{\boldsymbol{x}}_0, \boldsymbol{y}) \sim p_\theta(\hat{\boldsymbol{x}}_0, \boldsymbol{y}|\boldsymbol{x}_0^*)} \|\boldsymbol{y} - \boldsymbol{A}\hat{\boldsymbol{x}}_0\|^2 = \sigma_y^2 \mathrm{tr}(\boldsymbol{B}\boldsymbol{B}^\top) + \mathrm{tr}(\boldsymbol{A}\boldsymbol{\Sigma}_{\mathrm{post}}\boldsymbol{A}^\top),$$

where

$$\boldsymbol{\Sigma}_{\mathrm{post}} = \left( \frac{\boldsymbol{A}^\top \boldsymbol{A}}{\sigma_y^2} + \frac{\boldsymbol{I}_n}{\sigma_0^2} \right)^{-1}, \quad \boldsymbol{B} = \left( \boldsymbol{I}_m - \frac{\boldsymbol{A}\boldsymbol{\Sigma}_{\mathrm{post}}\boldsymbol{A}^\top}{\sigma_y^2} \right),$$

---

**Algorithm 1:** N-HMC

---

**Require:** # HMC iterations $K$, # leapfrog steps $L$, initial integration step size $\delta$, measurement noise schedule $\{\sigma_{y,k}\}$, $\boldsymbol{x}_T$, $\boldsymbol{y}$, $\mathcal{A}$, $\gamma$

1: **for** $k = 0$ to $K - 1$ **do**
2:     **repeat**
3:         $\boldsymbol{p} \sim \mathcal{N}(\boldsymbol{0}, \boldsymbol{I})$                                           `// Initial momentum`
4:         $\hat{\boldsymbol{x}}_0 = \text{DDIM}(\boldsymbol{x}_T)$
5:         $H_0 = \frac{1}{2}\|\boldsymbol{x}_T\|^2 + \frac{1}{2\sigma_{y,k}^2}\|\boldsymbol{y} - \mathcal{A}(\hat{\boldsymbol{x}}_0)\|^2 + \frac{1}{2}\boldsymbol{p}^\top \boldsymbol{p}$         `// Current Hamiltonian`
6:         $\boldsymbol{x}_T^* \leftarrow \boldsymbol{x}_T$                                 `// Initialize proposal` $\boldsymbol{x}_T$
7:         **for** $l = 0$ to $L - 1$ **do**
8:             $\boldsymbol{p} \leftarrow \boldsymbol{p} - \frac{\delta}{2}\left(\boldsymbol{x}_T^* + \frac{1}{2\sigma_{y,k}^2}\nabla_{\boldsymbol{x}_T^*}\|\boldsymbol{y} - \mathcal{A}(\hat{\boldsymbol{x}}_0^*)\|^2\right)$         `// Update momentum`
9:             $\boldsymbol{x}_T^* \leftarrow \boldsymbol{x}_T^* + \delta\boldsymbol{p}$                             `// Update` $\boldsymbol{x}_T^*$
10:           $\hat{\boldsymbol{x}}_0^* = \text{DDIM}(\boldsymbol{x}_T^*)$
11:             $\boldsymbol{p} \leftarrow \boldsymbol{p} - \frac{\delta}{2}\left(\boldsymbol{x}_T^* + \frac{1}{2\sigma_{y,k}^2}\nabla_{\boldsymbol{x}_T^*}\|\boldsymbol{y} - \mathcal{A}(\hat{\boldsymbol{x}}_0^*)\|^2\right)$         `// Update momentum`
12:         **end for**
13:         $H_1 = \frac{1}{2}\|\boldsymbol{x}_T^*\|^2 + \frac{1}{2\sigma_{y,k}^2}\|\boldsymbol{y} - \mathcal{A}(\hat{\boldsymbol{x}}_0^*)\|^2 + \frac{1}{2}\boldsymbol{p}^\top \boldsymbol{p}$       `// Proposal Hamiltonian`
14:         $u \sim \text{Unif}(0, 1)$
15:         **if** $u < \exp(H_0 - H_1)$ **then**
16:             Accept proposal
17:         **else**
18:             $\delta \leftarrow \gamma\delta$                                 `// Anneal step size` $\delta$
19:         **end if**
20:     **until** Proposal accepted
21:     $\boldsymbol{x}_T \leftarrow \boldsymbol{x}_T^*$                                   `// Accept the proposal`
22: **end for**
23: **return** $\boldsymbol{x}_T$

---

and $\boldsymbol{I}_m$ is the $m \times m$ identity matrix. $m$ denotes the dimension of $\boldsymbol{y}$.

The expected residual decomposes into two contributions: a noise-dependent term that appears only when measurement noise is present, and a second term that persists in all settings due to intrinsic uncertainty of prior diffusion models. In Corollary 1.1 below, we show that both terms behave in a way that yields a residual whose magnitude matches the true measurement noise.

**Corollary 1.1**   Under the assumptions of Proposition 1, if $\sigma_0/\sigma_y \ll 1$, the residual $\boldsymbol{y} - \mathcal{A}(\hat{\boldsymbol{x}}_0)$ satisfies

$$\mathbb{E}_{(\hat{\boldsymbol{x}}_0, \boldsymbol{y}) \sim p_\theta(\hat{\boldsymbol{x}}_0, \boldsymbol{y}|\boldsymbol{x}_0^*)}\|\boldsymbol{y} - \mathcal{A}(\boldsymbol{x}_0)\|^2 \to m\sigma_y^2.$$

In other words, the magnitude of residual aligns with the true known level of measurement noise, indicating that N-HMC remains robust and does not overfit to noise.

### 3.3 NOISE-ADAPTIVE NHMC

In practice, the type and level of measurement noise are often unknown, making the likelihood term $p(\boldsymbol{y} \mid \boldsymbol{x}_T)$ intractable. To address this, other methods usually have tunable hyperparameters that control the strength of the likelihood term or use task-specific early stopping criteria. Instead of this heuristic approach, we introduce a noise-adaptive sampling method, *NA-NHMC*, which extends N-HMC to the *unknown noise setting without any additional hyperparameter tuning*.

We treat the noise variance as a latent variable and adopt the Jeffreys prior, a principled noninformative choice due to its parameterization invariance. It is scale-invariant and represents maximal uncertainty about the noise level, making it appropriate when no prior information about $\sigma_y$ is available. It can also be viewed as the limiting case of an Inverse-Gamma prior $\sigma_y^2 \sim \text{Inv-}\Gamma(\alpha, \beta)$ as $\alpha, \beta \to 0$. The Inverse-Gamma distribution is the conjugate prior for the variance of a Gaussian likelihood. Proposition 2 characterizes the resulting behavior under additional assumptions.

$$p(\sigma_y^2) \sim \frac{1}{\sigma_y^2}.$$

**Proposition 2** Under the assumptions of Proposition 1 and that the pretrained diffusion model unconditionally generates images that lie on the high-quality manifold ($\sigma_0/\sigma_y \ll 1$), then the update rule of NA-NHMC follows:

$$\nabla_{\boldsymbol{x}_T} \log p(\boldsymbol{y}|\boldsymbol{x}_T)_{\text{NA-NHMC}} = -\frac{1}{2\sigma_y^2}\nabla_{\boldsymbol{x}_T}\|\boldsymbol{y} - \mathcal{A}(\mathcal{D}(\boldsymbol{x}_T))\|^2.$$

By marginalizing $\sigma_y^2$, the likelihood term becomes

$$p(\boldsymbol{y} \mid \boldsymbol{x}_T) = \int_0^\infty p(\boldsymbol{y} \mid \boldsymbol{x}_T, \sigma_y^2)\, p(\sigma_y^2)\, d\sigma_y^2 \tag{10}$$

$$\propto \left(\tfrac{1}{2}\big\|\boldsymbol{y} - \mathcal{A}(\mathcal{D}(\boldsymbol{x}_T))\big\|^2\right)^{-m/2}. \tag{11}$$

where $m$ denotes the dimensionality of the measurement space. The derivation of this expression is provided in Appendix A.1. Substituting this marginalized likelihood into the N-HMC framework yields our proposed noise-adaptive Algorithm 3. Proposition 2 below shows that, with an appropriate measurement noise prior, the likelihood term $\nabla_{\boldsymbol{x}_T} \log p(\boldsymbol{y}|\boldsymbol{x}_T)$ of NA-NHMC is identical to that of N-HMC (with known noise level).

Figure 2 demonstrates Proposition 2 in practice. All experiments use an identical setup across different noise levels, highlighting that our method does not require any hyperparameter tuning for a specific noise level. Despite the absence of such tunable parameters, the estimated standard deviation of the measurement noise, computed as $\|\boldsymbol{y} - \mathcal{A}(\hat{\boldsymbol{x}}_0)\|/\sqrt{m}$, closely matches the true, unknown noise level $\sigma_y$. This confirms that NA-NHMC effectively adapts to varying noise without specific tuning. The complete NA-NHMC is summarized in Algorithm 3.

The flexibility of NA-NHMC goes beyond noise-level robustness. Although NA-NHMC is formulated assuming Gaussian measurement noise, experiments with alternative noise types (Section 4.3) show that the method remains effective across other common noise distributions, demonstrating its broader robustness.

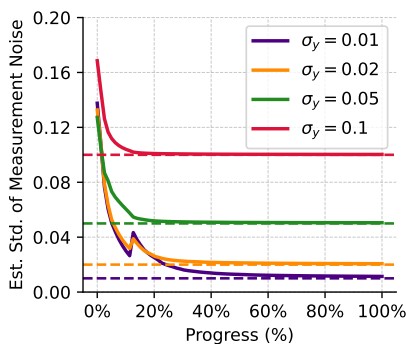

Figure 2: Gaussian deblur task on FFHQ ($256 \times 256$) with varying measurement noise levels $\sigma_y$. The estimated standard deviation of measurement noise $\|\boldsymbol{y} - \mathcal{A}(\hat{\boldsymbol{x}}_0)\|/\sqrt{m}$ demonstrates that our noise-adaptive method accurately recovers the true $\sigma_y$ (indicated by dashed line) without overfitting across different noise levels.

## 4 EXPERIMENTS

Following previous approaches (Wang et al., 2024) (Zhang et al., 2024), we evaluate our method on two datasets: FFHQ $256 \times 256$ (Karras et al., 2019) and ImageNet $256 \times 256$ (Deng et al., 2009), using 100 images from the validation set of each dataset. We utilize the same pretrained DM trained by Chung et al. (2023) for FFHQ and by Dhariwal & Nichol (2021) for ImageNet except for ReSample. For ReSample, we use a pretrained LDM by Rombach et al. (2022). All measurements are corrupted by additive Gaussian noise with standard deviation $\sigma_y$.

We compare our method against several representative baselines, including DiffPIR (Zhu et al., 2023), RED-diff (Mardani et al., 2023), DPS (Chung et al., 2023), DAPS (Zhang et al., 2024), ReSample (Song et al., 2024), SITCOM (Alkhouri et al., 2025b), and DMPlug (Wang et al., 2024). The implementation details for all the baseline methods are provided in Appendix A.5. We evaluate reconstruction quality using three standard metrics: peak signal-to-noise ratio (PSNR), structural similarity index measure (SSIM) (Wang et al., 2004), and Learned Perceptual Image Patch Similarity (LPIPS) (Zhang et al., 2018).

## 4.1 Experiment results

**Linear IPs.** We evaluate our approach on four linear inverse problems. For super-resolution tasks, we consider both $4\times$ and $16\times$ downsampling using $4 \times 4$ and $16 \times 16$ average pooling operations, respectively. We also examine random inpainting with $92\%$ of pixels randomly masked, and anisotropic Gaussian deblurring using blur kernels with standard deviations of 20 and 1 in orthogonal directions. The results for linear IPs are presented in Tables 10-13.

**Nonlinear IPs.** We further assess performance on three challenging nonlinear inverse problems. The first is nonlinear deblurring using encoded blur kernels from Tran et al. (2021). The second is phase retrieval, where only the Fourier magnitude is observed as measurements. Finally, we consider HDR reconstruction, which aims to recover images with a higher dynamic range by a factor of 2 from tone-mapped observations. The results for nonlinear IPs are presented in Tables 1, 2, 14, 15.

**Main Results.** Our method achieves comparable or superior performance across most tasks, as measured by PSNR and SSIM on both the FFHQ and ImageNet datasets. Notably, the improvement over SOTA methods is more pronounced for nonlinear tasks, which are substantially more challenging than linear IPs. Many existing SOTA approaches are MAP-based by design (e.g., ReSample, SITCOM, and DMPlug) or become MAP-like heuristically (e.g., DAPS). While these methods perform well in low-noise regimes, they often overfit when the noise level is higher. Since the noise levels used in our experiments ($\sigma_y = 0.05, 0.20$) exceed those commonly reported in prior work, our results further demonstrate that the proposed noise-adaptive method is more robust and consistently outperforms alternatives across most tasks and metrics without any hyperparameter tuning. Figure 4 contains visual examples for the nonlinear deblurring problem. See Appendix A.12 for more examples. A fundamental distinction lies in the generalization capability across diverse degradation conditions. Standard

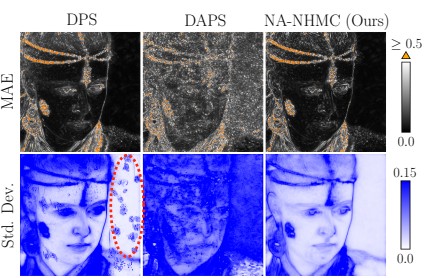

Figure 3: Comparative results are averaged over 100 independent runs. (Top) Mean absolute error (MAE) heapmaps. (Bottom) Standard deviation heatmaps across runs. Our method achieves the lowest standard deviation compared to DPS and DAPS, indicating reduced sensitivity to initialization.

guidance-based methods (e.g., DPS) inherently rely on manual hyperparameter calibration to balance measurement fidelity against the diffusion prior. As evidenced in Table 4, the optimal step size is highly task-dependent (ranging from $\zeta = 0.4$ to $\zeta = 10.0$), meaning a static configuration fails to generalize. In contrast, NA-NHMC derives its dynamics from the marginalized posterior, which effectively acts as an automatic gradient normalization mechanism. This structural advantage allows a single configuration to robustly generalize across varying tasks and noise levels without task-specific recalibration.

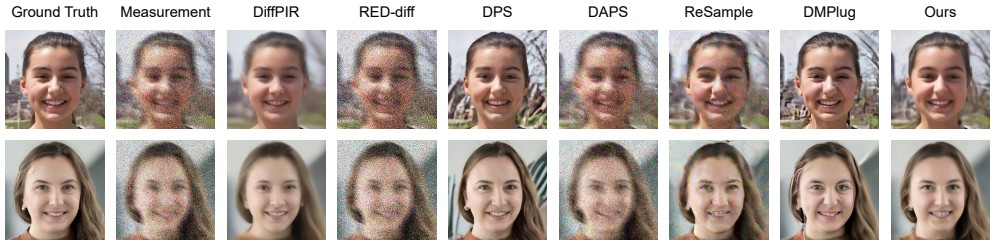

Figure 4: Nonlinear deblurring results on FFHQ ($256 \times 256$) dataset with $\sigma_y = 0.2$. Visual comparison across state-of-the-art methods shows our approach produces high-quality reconstructions with sharp details and minimal artifacts.

## 4.2 Highly ill-posed IPs: Phase Retrieval

Another challenge commonly faced by both MAP and sampling-based methods is becoming trapped in a local mode, particularly in highly multimodal IPs such as phase retrieval. Figure 5 illustrates this

issue. While DPS and DMPlug occasionally recover the correct solution, most initializations converge to spurious local modes and are thus counted as failures. In contrast, our method incorporates early exploration through $\sigma_y$ scheduling, making it more robust to initialization and substantially more likely to recover the global solution.

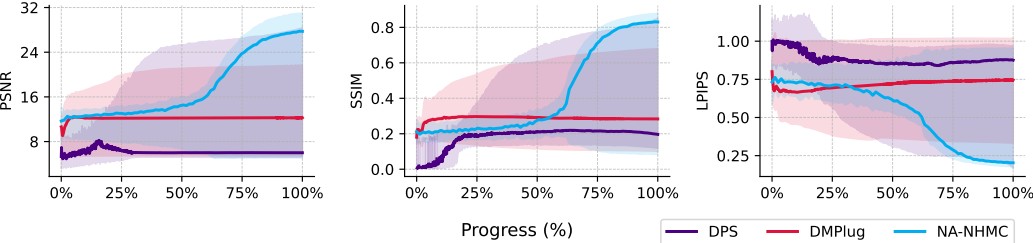

Figure 5: Phase retrieval task on FFHQ ($256 \times 256$) with $\sigma_y = 0.01$. Each curve shows the median performance, with shaded areas denoting the 5th–95th percentile interval. Our method successfully solves the IP at a much higher rate than DPS and DMPlug. This is due to the annealing schedule of $\sigma_y$ that allows for initial exploration of the noise space, resulting in a lower probability of being stuck on a local mode.

We quantify robustness to initialization using the standard deviation map in Figure 3. Our method achieves a mean absolute error (MAE) comparable to that of DPS, but with substantially lower pixel-wise standard deviation. While DPS can, on average, produce accurate reconstructions, its performance is sensitive to initialization and may introduce artifacts in both the face and background (red circle). In contrast, such artifacts never appear in any of the 100 runs with our method. Notably, despite exhibiting lower overall uncertainty, our method still assigns uncertainty in complex regions, which aligns with areas of high MAE (orange).

Table 1: Non-linear IPs Results on FFHQ ($256 \times 256$) with Gaussian Noise $\sigma_y = 0.05$. (**Bold**: best, underline: second best)

|  | Nonlinear Deblurring | | | Phase Retrieval | | | HDR Reconstruction | | |
|---|---|---|---|---|---|---|---|---|---|
|  | PSNR ↑ | SSIM ↑ | LPIPS ↓ | PSNR ↑ | SSIM ↑ | LPIPS ↓ | PSNR ↑ | SSIM ↑ | LPIPS ↓ |
| DiffPIR | 26.12 | 0.743 | 0.289 | 16.77 | 0.482 | 0.543 | 25.20 | 0.814 | 0.223 |
| RED-diff | 18.12 | 0.217 | 0.680 | 11.83 | 0.213 | 0.769 | 21.44 | 0.525 | 0.458 |
| DPS | 23.26 | 0.672 | 0.300 | 10.87 | 0.296 | 0.714 | 27.46 | **0.849** | **0.168** |
| DAPS | 27.00 | 0.736 | 0.283 | 18.52 | 0.414 | 0.528 | 26.03 | 0.758 | 0.259 |
| ReSample | 24.57 | 0.637 | 0.432 | 13.95 | 0.377 | 0.677 | 23.65 | 0.722 | 0.386 |
| SITCOM | 24.97 | 0.569 | 0.328 | 11.89 | 0.216 | 0.723 | 26.97 | 0.753 | 0.256 |
| DMPlug | 27.15 | 0.784 | 0.266 | - | - | - | 25.17 | 0.783 | 0.260 |
| **NA-NHMC (ours)** | **27.66** | **0.792** | **0.249** | **19.30** | **0.554** | **0.482** | **28.45** | 0.849 | 0.217 |

Table 2: Non-linear IPs on ImageNet ($256 \times 256$) with Gaussian Noise $\sigma_y = 0.05$. (**Bold**: best, underline: second best)

|  | Nonlinear Deblurring | | | HDR Reconstruction | | |
|---|---|---|---|---|---|---|
|  | PSNR ↑ | SSIM ↑ | LPIPS ↓ | PSNR ↑ | SSIM ↑ | LPIPS ↓ |
| DiffPIR | 24.24 | 0.638 | 0.381 | 23.29 | 0.730 | 0.273 |
| RED-diff | 17.94 | 0.244 | 0.623 | 20.98 | 0.524 | 0.415 |
| DPS | 17.60 | 0.427 | 0.482 | 25.31 | 0.763 | **0.248** |
| DAPS | 24.28 | 0.632 | 0.404 | 23.57 | 0.709 | 0.283 |
| SITCOM | 24.00 | 0.556 | 0.355 | 24.76 | 0.708 | 0.276 |
| DMPlug | 22.30 | 0.576 | 0.421 | 20.61 | 0.562 | 0.431 |
| **NA-NHMC (ours)** | **24.98** | **0.694** | **0.308** | **25.86** | **0.779** | 0.253 |

## 4.3 ROBUSTNESS TO UNKNOWN MEASUREMENT NOISE

In practice, the measurement noise may be unknown and its type may not be Gaussian. This can pose a problem as many methods require multiple hyperparameters that are tuned for a specific level

and type of measurement noise. In this section, we evaluate our method's robustness to unknown measurement noise on two tasks and two noise types: impulse and speckle. For impulse noise, each pixel in each channel is randomly replaced by $0$ or $1$ with a probability $p/2$ each, where $p \sim$ Unif$(0, 0.2)$. For speckle noise, the noise takes the form $y(1 + \epsilon)$, where $y$ is the measurement tensor and $\epsilon \sim$ Unif$(0, 0.4)$. Table 3 shows that our method achieves superior performance on most metrics while using the exact same hyperparameters as the Gaussian noise experiment. As illustrated in Figure 6, methods such as DiffPIR, which do not suffer from noise overfitting in the Gaussian setting, now struggle with impulse noise. In contrast, even under high noise levels, NA-NHMC remains robust, producing high-quality reconstructions.

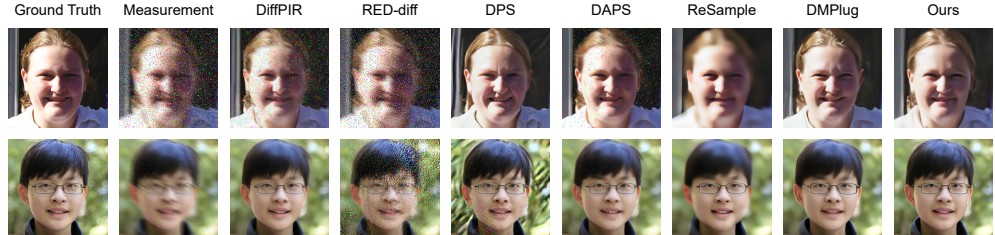

Figure 6: Nonlinear deblurring results on FFHQ $(256\times256)$ dataset under different noise conditions. (Top) Impulse noise. (Bottom) Speckle noise.

Table 3: The Performance Comparison with Different Types and Levels of Measurement Noise on FFHQ $(256 \times 256)$. (**Bold**: best, underline: second best)

| | Super Resolution ($\times4$) | | | | | | Nonlinear Deblurring | | | | | |
| | Impulse | | | Speckle | | | Impulse | | | Speckle | | |
| | PSNR ↑ | SSIM ↑ | LPIPS ↓ | PSNR ↑ | SSIM ↑ | LPIPS ↓ | PSNR ↑ | SSIM ↑ | LPIPS ↓ | PSNR ↑ | SSIM ↑ | LPIPS ↓ |
|---|---|---|---|---|---|---|---|---|---|---|---|---|
| DiffPIR | 19.54 | 0.492 | 0.549 | 25.91 | 0.733 | 0.324 | 21.00 | 0.402 | 0.526 | 25.96 | 0.731 | 0.299 |
| RED-diff | 15.15 | 0.341 | 0.692 | 21.81 | 0.481 | 0.516 | 13.82 | 0.109 | 0.781 | 18.59 | 0.269 | 0.650 |
| DPS | 21.99 | 0.581 | 0.395 | 27.00 | 0.761 | **0.246** | 21.64 | 0.595 | 0.322 | 23.42 | 0.678 | 0.302 |
| DAPS | 15.00 | 0.361 | 0.702 | 24.48 | 0.597 | 0.442 | 17.94 | 0.259 | 0.657 | 26.46 | 0.698 | 0.308 |
| ReSample | 22.98 | **0.639** | 0.483 | 26.17 | 0.733 | 0.387 | 22.74 | 0.616 | 0.471 | 24.64 | 0.692 | 0.409 |
| SITCOM | 16.56 | 0.392 | 0.628 | 23.16 | 0.600 | 0.425 | 17.92 | 0.259 | 0.612 | 26.49 | 0.667 | 0.295 |
| DMPlug | 19.52 | 0.358 | 0.562 | 26.79 | 0.689 | 0.336 | 23.79 | 0.662 | 0.335 | 25.82 | 0.740 | 0.308 |
| **NA-NHMC (ours)** | **23.42** | 0.631 | **0.382** | **27.36** | **0.768** | 0.290 | **24.16** | **0.677** | **0.319** | **27.97** | **0.796** | **0.253** |

## 5 CONCLUSION

In this work, we introduce N-HMC, a posterior sampler that operates in the noise space using reverse diffusion as a deterministic mapping from initial noise to a clean image, enabling posterior exploration while keeping proposals on the learned data manifold. The developed noise-adaptive variant, NA-NHMC, eliminates task-specific hyperparameter tuning by automatically adapting to unknown noise types and levels, which is a significant practical advantage over existing approaches. Theory establishes the correctness and efficiency of noise-space sampling, and experiments across diverse linear and nonlinear inverse problems on FFHQ and ImageNet show state-of-the-art reconstructions, robustness to initialization and noise, competitive runtimes with a few denoising steps, and uncertainty-aware estimates. The provided analysis and experiments also show that our method can mitigate measurement-consistency drift, noise overfitting, and local-mode collapse without relying on any task-specific hyperparameter tuning.

While NA-NHMC shows promising results, it incurs higher computational cost compared to other methods such as DPS due to HMC sampling. Additionally, its reliance on a small number of diffusion steps may limit its immediate applicability to more complex applications. Moreover, the high dimensionality of the posterior leads to long warmup phases before reaching stationarity. Future work could address these challenges by developing more efficient gradient estimation techniques and incorporating faster warmup strategies that relax the requirement of exact detailed balance.

## ACKNOWLEDGMENT

This research was supported by the Agency for Science, Technology and Research (A*STAR) under its RIE2025 Human Health and Potential (HHP) Industry Alignment Fund Pre-Positioning Programme (IAF-PP) (Grant No. H24J4a0145). We gratefully acknowledge the insightful comments and suggestions provided by the anonymous reviewers.

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

## ETHICS STATEMENT

We affirm adherence to the ICLR Code of Ethics. This work uses publicly available datasets and standard benchmarks; no human subjects, personal data, or sensitive attributes are collected or processed. We assessed potential risks (misuse, bias, fairness, privacy, and security) and found none beyond those commonly associated with generic image/model evaluation. All institutional and legal requirements were respected, and we disclose that we have no conflicts of interest or external sponsorship that could unduly influence the results.

## REPRODUCIBILITY STATEMENT

We provide implementation details, hyperparameters, and training/evaluation protocols in the main text and appendix; key equations and assumptions are stated with complete proofs in the appendix. All datasets and preprocessing steps are described and referenced; code and configuration files have been released in an repository to enable exact reproduction of results.

## A APPENDIX

### A.1 ROBUST-N-HMC DERIVATION

**Assumptions**

1. Measurement noise $\eta \in \mathbb{R}^m$ follows gaussian distribution with unknown $\sigma_y^2$:

$$p(\boldsymbol{y}|\boldsymbol{x}_T, \sigma_y^2) = \frac{1}{(2\pi\sigma_y^2)^{m/2}} \exp\left[-\frac{\|\boldsymbol{y} - \mathcal{A}(\mathcal{D}(\boldsymbol{x}_T))\|^2}{2\sigma_y^2}\right]. \tag{12}$$

2. $\sigma_y$ follows a Jeffreys prior distribution:

$$p(\sigma_y^2) \propto \frac{1}{\sigma_y^2}. \tag{13}$$

Marginalizing $\sigma_y^2$ yields

$$p(\boldsymbol{y}|\boldsymbol{x}_T) = \int_0^\infty p(\boldsymbol{y}|\boldsymbol{x}_T, \sigma_y^2)\, p(\sigma_y^2)\, d\sigma_y^2 \tag{14}$$

$$\propto \int_0^\infty \frac{1}{(2\pi\sigma_y^2)^{m/2}} \exp\left[-\frac{\|\boldsymbol{y} - \mathcal{A}(\mathcal{D}(\boldsymbol{x}_T))\|^2}{2\sigma_y^2}\right] \frac{1}{\sigma_y^2}\, d\sigma_y^2 \tag{15}$$

$$\propto \int_0^\infty (\sigma_y^2)^{-\frac{m}{2}-1} \exp\left[-\frac{(1/2)\|\boldsymbol{y} - \mathcal{A}(\mathcal{D}(\boldsymbol{x}_T))\|^2}{\sigma_y^2}\right] d\sigma_y^2 \tag{16}$$

$$\propto \left(\frac{1}{2}\|\boldsymbol{y} - \mathcal{A}(\mathcal{D}(\boldsymbol{x}_T))\|^2\right)^{-m/2}. \tag{17}$$

And then, we have

$$\log p(\boldsymbol{y}|\boldsymbol{x}_T) = \left(-\frac{m}{2}\right) \log\left(\frac{1}{2}\|\boldsymbol{y} - \mathcal{A}(\mathcal{D}(\boldsymbol{x}_T))\|^2\right). \tag{18}$$

### A.2 PROOFS

**Measurement model.** Let $\boldsymbol{x}_0^* \in \mathbb{R}^m$ denote the ground truth signal. The measurement is given by

$$\boldsymbol{y} = \mathcal{A}(\boldsymbol{x}_0^*) + \eta, \qquad \eta \sim \mathcal{N}(0, \sigma_y^2 \boldsymbol{I}_m), \tag{19}$$

where $\mathcal{A} : \mathbb{R}^n \to \mathbb{R}^m$ is the measurement operator and $\eta$ represents Gaussian measurement noise. In the following proofs, $\mathcal{A}$ is assumed to be approximately linear around $\boldsymbol{x}_0^*$. Thus, $\mathcal{A}(\boldsymbol{x}_0) = \boldsymbol{A}\boldsymbol{x}_0$.

**Generative model.** Consider the DDIM sampler defined by

$$\hat{\boldsymbol{x}}_0 = \mathcal{D}(\boldsymbol{x}_T), \qquad \boldsymbol{x}_T \sim \mathcal{N}(0, \boldsymbol{I}_n), \tag{20}$$

where $\mathcal{D}$ denotes the deterministic decoder via the diffusion model.

**Lemma 1.** Product of two Gaussian probability density functions (PDFs).

$$q_1(\boldsymbol{x}) = \mathcal{N}(\boldsymbol{x}; \mu_1, \boldsymbol{\Sigma}_1), \quad q_2(\boldsymbol{x}) = \mathcal{N}(\boldsymbol{x}; \mu_2, \boldsymbol{\Sigma}_2).$$

Then, the product of $q_1(\boldsymbol{x})$ and $q_2(\boldsymbol{x})$ is proportional to a Gaussian PDF $\mathcal{N}(\boldsymbol{x}, \mu, \boldsymbol{\Sigma})$, where

$$\mu = \boldsymbol{\Sigma}\left(\boldsymbol{\Sigma}_1^{-1}\mu_1 + \boldsymbol{\Sigma}_2^{-1}\mu_2\right), \quad \boldsymbol{\Sigma} = (\boldsymbol{\Sigma}_1^{-1} + \boldsymbol{\Sigma}_2^{-1})^{-1}.$$

**Lemma 2.** Bias-variance decomposition of a random variable $\boldsymbol{x}$ with mean $\mu$ and covariance matrix $\Sigma$, i.e.,

$$\mathbb{E}\big[\|\boldsymbol{x} - \boldsymbol{a}\|^2\big] = \|\boldsymbol{\mu} - \boldsymbol{a}\|^2 + \mathrm{tr}(\boldsymbol{\Sigma}).$$

**Proposition 1.** Assume that the distribution of the decoded sample $\boldsymbol{x}_0$ around the ground truth $\boldsymbol{x}_0^*$ is well-approximated by a Gaussian distribution $p_\theta(\hat{\boldsymbol{x}}_0) \approx \mathcal{N}(\hat{\boldsymbol{x}}_0; \boldsymbol{x}_0^*, \sigma_0^2 \boldsymbol{I}_n)$. Then, the residual $\boldsymbol{y} - \boldsymbol{A}\hat{\boldsymbol{x}}_0$ satisfies

$$\mathbb{E}_{(\hat{\boldsymbol{x}}_0, \boldsymbol{y}) \sim p_\theta(\hat{\boldsymbol{x}}_0, \boldsymbol{y} | \boldsymbol{x}_0^*)} \|\boldsymbol{y} - \boldsymbol{A}\hat{\boldsymbol{x}}_0\|^2 = \sigma_y^2 \mathrm{tr}(\boldsymbol{B}\boldsymbol{B}^\top) + \mathrm{tr}(\boldsymbol{A}\boldsymbol{\Sigma}_{\mathrm{post}}\boldsymbol{A}^\top),$$

where

$$\boldsymbol{\Sigma}_{\mathrm{post}} = \left(\frac{\boldsymbol{A}^\top \boldsymbol{A}}{\sigma_y^2} + \frac{\boldsymbol{I}_n}{\sigma_0^2}\right)^{-1}, \quad \boldsymbol{B} = \left(\boldsymbol{I}_m - \frac{\boldsymbol{A}\boldsymbol{\Sigma}_{\mathrm{post}}\boldsymbol{A}^\top}{\sigma_y^2}\right).$$

*Proof* First, consider the distribution of $\hat{\boldsymbol{x}}_0$ given fixed $\boldsymbol{x}_0^*$ and $\boldsymbol{y}$

$$p_\theta(\hat{\boldsymbol{x}}_0 | \boldsymbol{y}) \overset{(a)}{\propto} p_\theta(\boldsymbol{y} | \hat{\boldsymbol{x}}_0) p_\theta(\hat{\boldsymbol{x}}_0) \tag{21}$$

$$= \mathcal{N}(\boldsymbol{y}; \mathcal{A}(\hat{\boldsymbol{x}}_0), \sigma_y^2 \boldsymbol{I}_m) \, \mathcal{N}(\hat{\boldsymbol{x}}_0; \boldsymbol{x}_0^*, \sigma_0^2 \boldsymbol{I}_n) \tag{22}$$

$$\overset{(b)}{=} \mathcal{N}\big(\hat{\boldsymbol{x}}_0; (\boldsymbol{A}^\top \boldsymbol{A})^{-1} \boldsymbol{A}^\top \boldsymbol{y}, \sigma_y^2 (\boldsymbol{A}^\top \boldsymbol{A})^{-1}\big) \, \mathcal{N}(\hat{\boldsymbol{x}}_0; \boldsymbol{x}_0^*, \sigma_0^2 \boldsymbol{I}_n) \tag{23}$$

$$\overset{(c)}{=} \mathcal{N}\big(\hat{\boldsymbol{x}}_0; \boldsymbol{\mu}_{\mathrm{post}}(\boldsymbol{y}), \boldsymbol{\Sigma}_{\mathrm{post}}\big) \tag{24}$$

$$p_\theta(\boldsymbol{A}\hat{\boldsymbol{x}}_0 | \boldsymbol{y}) = \mathcal{N}\big(\hat{\boldsymbol{x}}_0; \boldsymbol{A}\boldsymbol{\mu}_{\mathrm{post}}(\boldsymbol{y}), \boldsymbol{A}\boldsymbol{\Sigma}_{\mathrm{post}}\boldsymbol{A}^\top\big), \tag{25}$$

where $(a)$ follows from Bayes' theorem. $(b)$ is from local linearity of $\mathcal{A}$. $(c)$ is the result of Lemma 1 with

$$\boldsymbol{\mu}_{\mathrm{post}}(\boldsymbol{y}) = \boldsymbol{\Sigma}_{\mathrm{post}} \left(\frac{\boldsymbol{A}^\top \boldsymbol{y}}{\sigma_y^2} + \frac{\boldsymbol{x}_0^*}{\sigma_0^2}\right), \quad \boldsymbol{\Sigma}_{\mathrm{post}} = \left(\frac{\boldsymbol{A}^\top \boldsymbol{A}}{\sigma_y^2} + \frac{\boldsymbol{I}_n}{\sigma_0^2}\right)^{-1}.$$

The expected squared residual conditioned on $\boldsymbol{y}$ is

$$\mathbb{E}_{\hat{\boldsymbol{x}}_0 \sim p_\theta(\hat{\boldsymbol{x}}_0 | \boldsymbol{y})} \|\boldsymbol{y} - \boldsymbol{A}\hat{\boldsymbol{x}}_0\|^2 = \|\boldsymbol{y} - \boldsymbol{A}\boldsymbol{\mu}_{\mathrm{post}}(\boldsymbol{y})\|^2 + \mathrm{tr}(\boldsymbol{A}\boldsymbol{\Sigma}_{\mathrm{post}}\boldsymbol{A}^\top), \tag{26}$$

which is the result of Lemma 2. Then, integrate over $\boldsymbol{y}$ conditioned on $\boldsymbol{x}_0^*$

$$\mathbb{E}_{\hat{\boldsymbol{x}}_0, \boldsymbol{y} \sim p_\theta(\hat{\boldsymbol{x}}_0, \boldsymbol{y} | \boldsymbol{x}_0^*)} \|\boldsymbol{y} - \boldsymbol{A}\hat{\boldsymbol{x}}_0\|^2 \tag{27}$$

$$= \mathbb{E}_{\boldsymbol{y} \sim q(\boldsymbol{y} | \boldsymbol{x}_0^*)} \big[\mathbb{E}_{\hat{\boldsymbol{x}}_0 \sim p_\theta(\hat{\boldsymbol{x}}_0 | \boldsymbol{y})} \|\boldsymbol{y} - \boldsymbol{A}\hat{\boldsymbol{x}}_0\|^2\big] \tag{28}$$

$$= \mathbb{E}_{\boldsymbol{y} \sim q(\boldsymbol{y} | \boldsymbol{x}_0^*)} \big[\|\boldsymbol{y} - \boldsymbol{A}\boldsymbol{\mu}_{\mathrm{post}}(\boldsymbol{y})\|^2 + \mathrm{tr}(\boldsymbol{A}\boldsymbol{\Sigma}_{\mathrm{post}}\boldsymbol{A}^\top)\big] \tag{29}$$

$$= \mathbb{E}_{\boldsymbol{y} \sim q(\boldsymbol{y} | \boldsymbol{x}_0^*)} \left[\left\|\left(\boldsymbol{I}_m - \frac{\boldsymbol{A}\boldsymbol{\Sigma}_{\mathrm{post}}\boldsymbol{A}^\top}{\sigma_y^2}\right)\boldsymbol{y} - \frac{\boldsymbol{A}\boldsymbol{\Sigma}_{\mathrm{post}}\boldsymbol{x}_0^*}{\sigma_0^2}\right\|^2\right] + \mathrm{tr}(\boldsymbol{A}\boldsymbol{\Sigma}_{\mathrm{post}}\boldsymbol{A}^\top) \tag{30}$$

$$\overset{(a)}{=} \left\|\left(\boldsymbol{I}_m - \frac{\boldsymbol{A}\boldsymbol{\Sigma}_{\mathrm{post}}\boldsymbol{A}^\top}{\sigma_y^2}\right)\boldsymbol{A}\boldsymbol{x}_0^* - \frac{\boldsymbol{A}\boldsymbol{\Sigma}_{\mathrm{post}}\boldsymbol{x}_0^*}{\sigma_0^2}\right\|^2 + \mathrm{tr}(\boldsymbol{B}(\sigma_y^2 \boldsymbol{I}_m)\boldsymbol{B}^\top) + \mathrm{tr}(\boldsymbol{A}\boldsymbol{\Sigma}_{\mathrm{post}}\boldsymbol{A}^\top) \tag{31}$$

$$= \sigma_y^2 \mathrm{tr}(\boldsymbol{B}\boldsymbol{B}^\top) + \mathrm{tr}(\boldsymbol{A}\boldsymbol{\Sigma}_{\mathrm{post}}\boldsymbol{A}^\top), \tag{32}$$

where $(a)$ is the result of Lemma 2, and $\boldsymbol{B} = \boldsymbol{I}_m - \frac{\boldsymbol{A}\boldsymbol{\Sigma}_{\mathrm{post}}\boldsymbol{A}^\top}{\sigma_y^2}$.

**Corollary 1.1** Under the assumptions of Proposition 1, if $\sigma_0 / \sigma_y \ll 1$, the residual $\boldsymbol{y} - \mathcal{A}(\hat{\boldsymbol{x}}_0)$ satisfies

$$\mathbb{E}_{(\hat{\boldsymbol{x}}_0, \boldsymbol{y}) \sim p_\theta(\hat{\boldsymbol{x}}_0, \boldsymbol{y} | \boldsymbol{x}_0^*)} \|\boldsymbol{y} - \mathcal{A}(\boldsymbol{x}_0)\|^2 \to m\sigma_y^2.$$

*Proof* If $\sigma_0 / \sigma_y \ll 1$,

$$\boldsymbol{\Sigma}_{\mathrm{post}} = \left(\frac{\boldsymbol{A}^\top \boldsymbol{A}}{\sigma_y^2} + \frac{\boldsymbol{I}_n}{\sigma_0^2}\right)^{-1} = \sigma_0^2 \left(\frac{\sigma_0^2 \boldsymbol{A}^\top \boldsymbol{A}}{\sigma_y^2} + \boldsymbol{I}_n\right)^{-1} \to \sigma_0^2 \boldsymbol{I}_n \tag{33}$$

$$\boldsymbol{B} = \boldsymbol{I}_m - \frac{\boldsymbol{A}\boldsymbol{\Sigma}_{\mathrm{post}}\boldsymbol{A}^\top}{\sigma_y^2} \to \boldsymbol{I}_m - \frac{\sigma_0^2 \boldsymbol{A}\boldsymbol{A}^\top}{\sigma_y^2} \to \boldsymbol{I}_m \tag{34}$$

$$\mathbb{E}\|\boldsymbol{y} - \mathcal{A}(\boldsymbol{x}_0)\|^2 = \sigma_y^2 \mathrm{tr}(\boldsymbol{B}\boldsymbol{B}^\top) + \mathrm{tr}(\boldsymbol{A}\boldsymbol{\Sigma}_{\mathrm{post}}\boldsymbol{A}^\top) \to m\sigma_y^2 \tag{35}$$

**Proposition 2** Under the assumptions of Proposition 1 and that the pre-trained diffusion model unconditionally generates images that lie on the high quality manifold ($\sigma_0/\sigma_y \ll 1$), then the update rule of NA-NHMC follows:

$$\nabla_{\boldsymbol{x}_T} \log p(\boldsymbol{y}|\boldsymbol{x}_T)_{\text{NA-NHMC}} = -\frac{1}{2\sigma_y^2} \nabla_{\boldsymbol{x}_T} \|\boldsymbol{y} - \mathcal{A}(\mathcal{D}(\boldsymbol{x}_T))\|^2$$

*Proof* We can consider the likelihood term of NA-NHMC

$$\log p(\boldsymbol{y}|\boldsymbol{x}_T) = \left(-\frac{m}{2}\right) \log \left(\frac{1}{2}\|\boldsymbol{y} - \mathcal{A}(\mathcal{D}(\boldsymbol{x}_T))\|^2\right) \tag{36}$$

$$\nabla_{\boldsymbol{x}_T} \log p(\boldsymbol{y}|\boldsymbol{x}_T) = \left(-\frac{m}{2\|\boldsymbol{y} - \mathcal{A}(\mathcal{D}(\boldsymbol{x}_T))\|^2}\right) \nabla_{\boldsymbol{x}_T} \|\boldsymbol{y} - \mathcal{A}(\mathcal{D}(\boldsymbol{x}_T))\|^2 \tag{37}$$

$$= \left(-\frac{1}{2\sigma_y^2}\right) \nabla_{\boldsymbol{x}_T} \|\boldsymbol{y} - \mathcal{A}(\mathcal{D}(\boldsymbol{x}_T))\|^2, \tag{38}$$

which follows from Corollary 1.1. This likelihood term is exactly the same as that of N-HMC, where the true noise level $\sigma_y$ is known.

## A.3 PSEUDOCODE OF UNCONDITIONAL DDIM

The DDIM method (Song et al., 2022a) we used in our experiment follows the Algorithm 2 below:

---

**Algorithm 2: DDIM**

---

**Require:** # diffusion steps $T$, diffusion model $\boldsymbol{s}_\theta$, initial seed $\boldsymbol{x}_T$
1: **for** $t = T - 1$ to 0 **do**
2:    $\hat{\boldsymbol{\epsilon}}_{t+1} = \boldsymbol{s}_\theta(\boldsymbol{x}_{t+1}, t+1)$                                   // Compute the score
3:    $\hat{\boldsymbol{x}}_0(\boldsymbol{x}_{t+1}) = \frac{1}{\sqrt{\overline{\alpha}_{t+1}}}\left(\boldsymbol{x}_{t+1} - \sqrt{1 - \overline{\alpha}_{t+1}}\hat{\boldsymbol{\epsilon}}_{t+1}\right)$ // Predict $\hat{\boldsymbol{z}}_0$ with Tweedie's formula
4:    $\hat{\boldsymbol{x}}_t = \sqrt{\overline{\alpha}_t}\hat{\boldsymbol{x}}_0(\boldsymbol{x}_{t+1}) + \sqrt{1 - \overline{\alpha}_t}\hat{\boldsymbol{\epsilon}}_{t+1}$                // Unconditional DDIM step
5: **end for**
6: **return** $\boldsymbol{x}$

---

## A.4 PSEUDOCODE OF THE NOISE-ADAPTIVE NHMC

Following the reasoning in Proposition 2, we assume an uninformative prior on $\sigma_y$. Under this assumption, the likelihood term can be written as

$$\log p(\boldsymbol{y}|\boldsymbol{x}_T) = \left(-\frac{m}{2}\right) \log \left(\frac{1}{2}\|\boldsymbol{y} - \mathcal{A}(\mathcal{D}(\boldsymbol{x}_T))\|^2\right).$$

where $m = \dim(\boldsymbol{y})$. The factor $1/2$ inside the logarithm can be omitted for the Hamiltonian and gradient computations. The corresponding gradient is

$$\nabla_{\boldsymbol{x}_T} \log p(\boldsymbol{y}|\boldsymbol{x}_T) = \left(-\frac{m}{2\|\boldsymbol{y} - \mathcal{A}(\mathcal{D}(\boldsymbol{x}_T))\|^2}\right) \nabla_{\boldsymbol{x}_T} \|\boldsymbol{y} - \mathcal{A}(\mathcal{D}(\boldsymbol{x}_T))\|^2$$

## A.5 IMPLEMENTATION DETAILS FOR BASELINE METHODS

**DiffPIR**

Number of diffusion steps: 100
Number of optimization steps: 50

We follow the recommended hyperparameter $\eta = 1.0$ and $\lambda = 7.0$ from Zhang et al. (2025). The learning rate of the schedule-free AdamW optimizer is set to 0.1.

---

**Algorithm 3:** NA-NHMC

---

**Require:** # HMC iterations $K$, # leapfrog steps $L$, initial integration step size $\delta$, $\boldsymbol{x}_T$, $\boldsymbol{y}$, $\mathcal{A}$, $\gamma$

1: **for** $k = 0$ to $K - 1$ **do**
2:    **repeat**
3:       $\boldsymbol{p} \sim \mathcal{N}(\boldsymbol{0}, \boldsymbol{I})$                         `// Initial momentum`
4:       $\hat{\boldsymbol{x}}_0 = \text{DDIM}(\boldsymbol{x}_T)$
5:       $H_0 = \frac{1}{2}\|\boldsymbol{x}_T\|^2 + \frac{m}{2}\log\left(\|\boldsymbol{y} - \mathcal{A}(\hat{\boldsymbol{x}}_0)\|^2\right) + \frac{1}{2}\boldsymbol{p}^\top\boldsymbol{p}$      `// Current Hamiltonian`
6:       $\boldsymbol{x}_T^* \leftarrow \boldsymbol{x}_T$                    `// Initialize proposal `$\boldsymbol{x}_T$
7:       **for** $l = 0$ to $L - 1$ **do**
8:          $\boldsymbol{p} \leftarrow \boldsymbol{p} - \frac{\delta}{2}\left(\boldsymbol{x}_T^* + \frac{m}{2\|\boldsymbol{y} - \mathcal{A}(\hat{\boldsymbol{x}}_0^*)\|^2}\nabla_{\boldsymbol{x}_T^*}\|\boldsymbol{y} - \mathcal{A}(\hat{\boldsymbol{x}}_0^*)\|^2\right)$    `// Update momentum`
9:          $\boldsymbol{x}_T^* \leftarrow \boldsymbol{x}_T^* + \delta\boldsymbol{p}$                  `// Update `$\boldsymbol{x}_T^*$
10:         $\hat{\boldsymbol{x}}_0^* = \text{DDIM}(\boldsymbol{x}_T^*)$
11:          $\boldsymbol{p} \leftarrow \boldsymbol{p} - \frac{\delta}{2}\left(\boldsymbol{x}_T^* + \frac{m}{2\|\boldsymbol{y} - \mathcal{A}(\hat{\boldsymbol{x}}_0^*)\|^2}\nabla_{\boldsymbol{x}_T^*}\|\boldsymbol{y} - \mathcal{A}(\hat{\boldsymbol{x}}_0^*)\|^2\right)$    `// Update momentum`
12:       **end for**
13:       $H_1 = \frac{1}{2}\|\boldsymbol{x}_T^*\|^2 + \frac{m}{2}\log\left(\|\boldsymbol{y} - \mathcal{A}(\hat{\boldsymbol{x}}_0^*)\|^2\right) + \frac{1}{2}\boldsymbol{p}^\top\boldsymbol{p}$      `// Proposal Hamiltonian`
14:       $u \sim \text{Unif}(0, 1)$
15:       **if** $u < \exp(H_0 - H_1)$ **then**
16:          Accept proposal
17:       **else**
18:          $\delta \leftarrow \gamma\delta$                   `// Anneal step size `$\delta$
19:       **end if**
20:    **until** Proposal accepted
21:    $\boldsymbol{x}_T \leftarrow \boldsymbol{x}_T^*$                     `// Accept the proposal`
22: **end for**
23: **return** $\boldsymbol{x}_T$

---

**RED-diff**

Number of optimization steps: 1000

We follow the recommended the hyperparameter $\lambda = 0.25$ and an Adam optimizer with $lr = 0.5$ as in Zhang et al. (2025)

**DPS**

Number of diffusion steps: 1000

We follow the learning rate form in Chung et al. (2023) with $\zeta_i$ adjusted for different tasks, as shown in Table 4.

Table 4: Tuned learning rate $\zeta_i$ for DPS

| | SR ($\times 4$) | SR ($\times 16$) | Inpainting (92%) | Gaussian Deblurring | Nonlinear Deblurring | Phase Retrieval | HDR |
|---|---|---|---|---|---|---|---|
| FFHQ | 1.0 | 0.6 | 1.0 | 1.0 | 1.0 | 0.4 | 1.0 |
| ImageNet | 1.0 | 0.6 | 1.0 | 0.4 | 0.5 | - | 1.0 |

**DAPS**

Number of diffusion steps: 250

Number of ODE solver steps: 4

We follow the hyperparameter settings of Zhang et al. (2024), as listed in Table 5, and adopt their heuristic $\sigma_y = 0.01$ in place of the actual value. $\delta = 0.01$ for all tasks. The number of MCMC sampling steps $N = 100$ for FFHQ ($256 \times 256$) and $N = 40$ for ImageNet ($256 \times 256$). Otherwise, the hyperparameter for each task is the same for both datasets.

**ReSample**

Number of diffusion steps: 500

Table 5: $\eta_0$ for DAPS

| SR ($\times$4) | SR ($\times$16) | Inpainting (92%) | Gaussian Deblurring | Nonlinear Deblurring | Phase Retrieval | HDR |
|---|---|---|---|---|---|---|
| 1e-4 | 1e-4 | 1e-4 | 1e-4 | 5e-5 | 5e-5 | 2e-5 |

For both noise levels ($\sigma_y = 0.05, 0.20$) tested in this paper, the recommended optimization steps lead to overfitting to noise and poor performance. Instead, we used 50 steps for pixel optimization and 25 steps for latent optimization.

**SITCOM**

Number of diffusion steps: 20

We follow the hyperparameter settings of Alkhouri et al. (2025a), as listed in Table 6. The stopping criterion $\delta$ for $\sigma_y \in \{0.05, 0.2\}$ is chosen as $0.051\sqrt{m}$ and $0.201\sqrt{m}$ respectively, with $m$ denoting the dimension of $y$.

Table 6: Optimization Steps $K$ for SITCOM

| SR ($\times$4) | SR ($\times$16) | Inpainting (92%) | Gaussian Deblurring | Nonlinear Deblurring | Phase Retrieval | HDR |
|---|---|---|---|---|---|---|
| 20 | 20 | 30 | 30 | 30 | 30 | 40 |

**DMPlug**

Number of diffusion steps: 3

$t = [250, 500, 750]$

We set the Adam optimizer learning rate to 0.01. We follow the recommended stopping criteria in Wang et al. (2024). For linear tasks, we use a window size = 10, patience = 100, and a maximum iterations = 5000. For nonlinear tasks, we use the window size = 50 and patience = 300 with maximum iterations = 10000.

## A.6 IMPLEMENTATION DETAILS FOR OUR METHOD

Number of diffusion steps: 2

$t = [375, 750]$

We implement NA-NHMC with the same hyperparameter configuration $L = 20, \delta_0 = 0.05, \gamma = 0.95$ for both datasets. For all tasks except phase retrieval, an initial step size $\delta_0 = 0.05$ and an annealing schedule $\sigma_{y,k} = 0.5 + 2(1 - k/10)$ is applied during the first 10 HMC iterations, after which, the noise-adaptive scheme (Algorithm 3) is used.

For phase retrieval, we use initial step size $\delta_0 = 0.2$ and keep other hyperparameters unchanged. An annealing schedule $\sigma_{y,k} = 1.0 + 20\sqrt{1 - k/50}$ is applied during the first 50 HMC iterations, after which, the noise-adaptive scheme (Algorithm 3) is used.

These annealing schedules are chosen to encourage sufficient exploration of the posterior in the early stage. Our results are not sensitive to the exact schedule: using a slower schedule does not degrade performance.

## A.7 HYPERPARAMETER SENSITIVITY ANALYSIS

We evaluate several hyperparameter choices for the Hamiltonian Monte Carlo sampler on the super-resolution ($\times$4) inverse problem using the FFHQ $256 \times 256$ dataset. When varying a particular hyperparameter, all remaining hyperparameters are kept at their default values used in all other experiments. For different choices of the number of leapfrog steps $L$, we also adjust the number of HMC iterations to ensure that each setting uses the same amount of computational resources. The

resulting performance is summarized in the tables below. The results indicate that the performance of NA-NHMC exhibits little sensitivity to the choice of hyperparameters.

Table 7: Different step sizes $\epsilon$ for Super Resolution ($\times 4$) on FFHQ ($256 \times 256$) with Gaussian Noise $\sigma_y = 0.05$. (**Bold**: best)

|  | 0.02 | 0.05 | 0.10 | 0.15 | 0.20 |
|---|---|---|---|---|---|
| PSNR ↑ | 27.12 | 27.29 | **27.31** | **27.31** | **27.31** |
| SSIM ↑ | 0.745 | 0.770 | 0.771 | 0.771 | **0.772** |
| LPIPS ↓ | 0.299 | 0.291 | 0.288 | 0.288 | **0.286** |

Table 8: Different number of leapfrog steps $L$ for Super Resolution ($\times 4$) on FFHQ ($256 \times 256$) with Gaussian Noise $\sigma_y = 0.05$. (**Bold**: best)

|  | 10 | 15 | 20 | 25 | 30 |
|---|---|---|---|---|---|
| PSNR ↑ | 26.86 | 27.18 | 27.29 | **27.36** | 27.34 |
| SSIM ↑ | 0.749 | 0.765 | 0.770 | 0.771 | **0.777** |
| LPIPS ↓ | 0.318 | 0.299 | 0.291 | 0.286 | **0.281** |

Table 9: Step size decay factor $\gamma$ for Super Resolution ($\times 4$) on FFHQ ($256 \times 256$) with Gaussian Noise $\sigma_y = 0.05$. (**Bold**: best)

|  | 0.91 | 0.93 | 0.95 | 0.97 | 0.99 |
|---|---|---|---|---|---|
| PSNR ↑ | 27.29 | 27.31 | 27.29 | **27.31** | 27.30 |
| SSIM ↑ | 0.770 | 0.769 | 0.770 | **0.771** | 0.770 |
| LPIPS ↓ | 0.291 | 0.289 | 0.291 | **0.288** | 0.289 |

## A.8 ADDITIONAL EXPERIMENT RESULTS

**Linear IPs results**

All experiments in Section 3.3 are repeated with **a higher level of Gaussian measurement noise** ($\sigma_y = 0.20$). The results are shown below.

## A.9 ABLATION STUDIES

**Number of HMC iterations**

Sampling with HMC requires a warmup phase, since the initial noise $x_T$ may be far from the solution. As shown in Figure 7, the quality of sampled images improves monotonically with the number of iterations, as expected. Performance begins to plateau after roughly 120 iterations. Unlike MAP-based methods such as ReSample (Song et al., 2024) and DMPlug (Wang et al., 2024), it does not deteriorate beyond this point. This stability provides evidence that the prior term acts as an effective regularizer, preventing overfitting.

**Number of diffusion steps and memory usage**

In this section, we evaluate NA-NHMC with varying numbers of diffusion steps, using fixed parameters $K = 80$ and $L = 20$. Both runtime (in seconds) and memory usage (in GB) increase linearly with the number of steps. We ran all experiments on NVIDIA H200 GPU. The baseline cost is 90 seconds and 3.63 GB for two steps, with each additional step adding roughly 45 seconds and 1.84 GB. The quantitative evaluations are shown in Figure 8. While three diffusion steps achieve the highest PSNR and lowest LPIPS, the improvement over two steps is marginal. To avoid incurring roughly 50% additional runtime and memory overhead, we use two diffusion steps in all experiments.

Note that performance appears to decline when using more than three diffusion steps. This effect arises because the sampler converges more slowly to its stationary distribution as the number of

Table 10: Linear IPs on FFHQ (256 × 256) with Gaussian Noise $\sigma_y = 0.05$. (**Bold**: best, underline: second best)

| | Super Resolution (×4) | | | Super Resolution (×16) | | | Random Inpainting (92%) | | | Gaussian Deblurring | | |
|---|---|---|---|---|---|---|---|---|---|---|---|---|
| | PSNR ↑ | SSIM ↑ | LPIPS ↓ | PSNR ↑ | SSIM ↑ | LPIPS ↓ | PSNR ↑ | SSIM ↑ | LPIPS ↓ | PSNR ↑ | SSIM ↑ | LPIPS ↓ |
| DiffPIR | 25.96 | 0.735 | 0.322 | 19.84 | 0.541 | 0.444 | 20.93 | 0.595 | 0.405 | 27.48 | 0.778 | 0.287 |
| RED-diff | 21.58 | 0.390 | 0.602 | 17.60 | 0.391 | 0.567 | 23.70 | 0.651 | 0.344 | 17.07 | 0.213 | 0.692 |
| DPS | 26.84 | 0.762 | **0.239** | 20.06 | 0.522 | **0.380** | 25.74 | 0.745 | **0.245** | 26.88 | 0.761 | **0.234** |
| DAPS | 24.58 | 0.559 | 0.514 | 17.28 | 0.420 | 0.541 | 25.68 | 0.685 | 0.331 | 23.34 | 0.478 | 0.474 |
| ReSample | 26.18 | 0.737 | 0.382 | 20.01 | 0.532 | 0.576 | 24.12 | 0.599 | 0.442 | 25.98 | 0.728 | 0.385 |
| SITCOM | **27.35** | **0.787** | 0.268 | 20.82 | **0.574** | 0.400 | 26.56 | **0.785** | 0.266 | 27.94 | 0.796 | 0.266 |
| DMPlug | 26.73 | 0.697 | 0.321 | 17.42 | 0.280 | 0.607 | 26.15 | 0.769 | 0.270 | 27.81 | 0.769 | 0.289 |
| **NA-NHMC (ours)** | 27.29 | 0.770 | 0.291 | **20.85** | 0.531 | 0.452 | **26.72** | **0.785** | 0.268 | **28.36** | **0.798** | 0.259 |

Table 11: Linear IPs ImageNet (256 × 256) with Gaussian Noise $\sigma_y = 0.05$. (**Bold**: best, underline: second best)

| | Super Resolution (×4) | | | Super Resolution (×16) | | | Random Inpainting (92%) | | | Gaussian Deblurring | | |
|---|---|---|---|---|---|---|---|---|---|---|---|---|
| | PSNR ↑ | SSIM ↑ | LPIPS ↓ | PSNR ↑ | SSIM ↑ | LPIPS ↓ | PSNR ↑ | SSIM ↑ | LPIPS ↓ | PSNR ↑ | SSIM ↑ | LPIPS ↓ |
| DiffPIR | 23.99 | 0.626 | 0.426 | 18.48 | 0.387 | 0.626 | 19.30 | 0.443 | 0.583 | 25.24 | 0.678 | 0.378 |
| RED-diff | 17.67 | 0.266 | 0.613 | 12.45 | 0.151 | 0.726 | 17.25 | 0.360 | 0.541 | 13.99 | 0.170 | 0.688 |
| DPS | 23.36 | 0.623 | 0.345 | 17.15 | 0.339 | **0.524** | 22.31 | 0.593 | 0.347 | 22.55 | 0.555 | 0.401 |
| DAPS | 23.86 | 0.568 | 0.461 | 14.29 | 0.139 | 0.753 | 23.23 | 0.585 | 0.432 | 24.52 | 0.558 | 0.423 |
| SITCOM | 24.93 | **0.684** | 0.318 | 18.58 | **0.404** | 0.525 | 23.89 | **0.684** | 0.320 | 25.63 | **0.712** | **0.311** |
| DMPlug | 24.52 | 0.667 | 0.378 | 16.74 | 0.311 | 0.590 | 23.49 | 0.668 | 0.358 | 23.55 | 0.605 | 0.433 |
| **NA-NHMC (ours)** | **24.99** | 0.665 | 0.355 | **19.09** | **0.396** | 0.580 | **24.10** | 0.676 | 0.324 | **25.76** | 0.699 | 0.327 |

Table 12: Linear IPs on FFHQ (256 × 256) with Gaussian Noise $\sigma_y = 0.20$. (**Bold**: best, underline: second best)

| | Super Resolution (×4) | | | Super Resolution (×16) | | | Random Inpainting (92%) | | | Gaussian Deblurring | | |
|---|---|---|---|---|---|---|---|---|---|---|---|---|
| | PSNR ↑ | SSIM ↑ | LPIPS ↓ | PSNR ↑ | SSIM ↑ | LPIPS ↓ | PSNR ↑ | SSIM ↑ | LPIPS ↓ | PSNR ↑ | SSIM ↑ | LPIPS ↓ |
| DiffPIR | 21.22 | 0.591 | 0.417 | 16.09 | 0.420 | 0.551 | 17.83 | 0.470 | 0.509 | 24.29 | 0.683 | 0.355 |
| RED-diff | 12.64 | 0.101 | 0.824 | 11.49 | 0.152 | 0.799 | 15.33 | 0.168 | 0.731 | 8.47 | 0.037 | 0.868 |
| DPS | 21.80 | 0.556 | 0.385 | 16.13 | 0.377 | 0.507 | 21.60 | 0.531 | 0.404 | 24.45 | 0.678 | **0.290** |
| DAPS | 13.48 | 0.121 | 0.792 | 17.08 | 0.420 | 0.549 | 20.64 | 0.353 | 0.587 | 8.12 | 0.031 | 0.862 |
| ReSample | 22.95 | 0.632 | 0.501 | **17.93** | 0.468 | 0.661 | 22.62 | 0.615 | 0.535 | 24.60 | 0.682 | 0.438 |
| SITCOM | 23.04 | **0.647** | **0.362** | 17.49 | **0.469** | **0.495** | 23.23 | 0.653 | 0.359 | 24.97 | 0.709 | 0.323 |
| DMPlug | 15.95 | 0.140 | 0.706 | 11.69 | 0.093 | 0.773 | 19.65 | 0.347 | 0.564 | 17.33 | 0.181 | 0.660 |
| **NA-NHMC (ours)** | **23.29** | 0.636 | 0.391 | 17.36 | 0.393 | 0.552 | **23.69** | **0.670** | 0.365 | **25.57** | **0.710** | 0.327 |

Table 13: Linear IPs on ImageNet (256 × 256) with Gaussian Noise $\sigma_y = 0.20$. (**Bold**: best, underline: second best)

| | Super Resolution (×4) | | | Super Resolution (×16) | | | Random Inpainting (92%) | | | Gaussian Deblurring | | |
|---|---|---|---|---|---|---|---|---|---|---|---|---|
| | PSNR ↑ | SSIM ↑ | LPIPS ↓ | PSNR ↑ | SSIM ↑ | LPIPS ↓ | PSNR ↑ | SSIM ↑ | LPIPS ↓ | PSNR ↑ | SSIM ↑ | LPIPS ↓ |
| DiffPIR | 19.90 | 0.448 | 0.577 | **18.48** | **0.387** | 0.626 | 16.91 | 0.295 | 0.684 | 22.38 | 0.550 | 0.493 |
| RED-diff | 11.35 | 0.097 | 0.776 | 9.29 | 0.098 | 0.811 | 11.37 | 0.079 | 0.782 | 8.08 | 0.043 | 0.817 |
| DPS | 19.29 | 0.406 | 0.481 | 11.25 | 0.145 | 0.728 | 18.64 | 0.361 | 0.505 | 20.33 | 0.455 | 0.468 |
| DAPS | 13.71 | 0.151 | 0.760 | 14.27 | 0.139 | 0.755 | 18.89 | 0.248 | 0.598 | 9.06 | 0.060 | 0.789 |
| SITCOM | 20.87 | 0.490 | **0.458** | 16.16 | 0.325 | 0.628 | 20.90 | 0.494 | 0.457 | 22.73 | 0.584 | **0.402** |
| DMPlug | 18.58 | 0.412 | 0.500 | 9.53 | 0.125 | 0.782 | 19.15 | 0.447 | 0.474 | 23.05 | 0.591 | 0.423 |
| **NA-NHMC (ours)** | **21.53** | **0.510** | 0.492 | 16.43 | 0.276 | 0.675 | **21.80** | **0.536** | **0.456** | **23.45** | **0.597** | 0.405 |

diffusion steps increases. While increasing the number of HMC iterations could offset this effect, it would further amplify runtime costs to an impractical level.

**Diffusion schedule**

The pre-trained DMs used in this paper have 1000 diffusion steps. While other methods usually use evenly-spaced schedule with the first step bing pure Gaussian noise ($\overline{a}_t = 0$), we found this choice to be numerically unstable for our few-step setting. Since we are using two steps for unconditional DDIM, the natural choice is to use timesteps in the middle. Thus, we choose $t = [375, 750]$, which is spread evenly and avoids numerical stability. Table 16 confirms that this schedule yields superior performance in PSNR and SSIM while being close to optimal for LPIPS. We adopt this diffusion schedule for all main experiments.

Table 14: Non-linear IPs on FFHQ ($256 \times 256$) with Gaussian Noise $\sigma_y = 0.20$. (**Bold**: best, underline: second best)

| | Nonlinear Deblurring | | | Phase Retrieval | | | HDR Reconstruction | | |
|---|---|---|---|---|---|---|---|---|---|
| | PSNR ↑ | SSIM ↑ | LPIPS ↓ | PSNR ↑ | SSIM ↑ | LPIPS ↓ | PSNR ↑ | SSIM ↑ | LPIPS ↓ |
| DiffPIR | 23.34 | 0.641 | 0.374 | **16.76** | **0.482** | **0.543** | 21.85 | 0.694 | 0.344 |
| RED-diff | 12.85 | 0.063 | 0.816 | 10.07 | 0.061 | 0.855 | 16.73 | 0.222 | 0.649 |
| DPS | 22.83 | 0.643 | **0.307** | 10.60 | 0.267 | 0.701 | 24.92 | 0.703 | 0.321 |
| DAPS | 17.38 | 0.154 | 0.728 | 12.93 | 0.103 | 0.797 | 18.04 | 0.299 | 0.607 |
| ReSample | 23.30 | 0.635 | 0.477 | 12.51 | 0.335 | 0.712 | 22.51 | 0.677 | 0.428 |
| SITCOM | 16.26 | 0.173 | 0.656 | 10.19 | 0.082 | 0.810 | 20.11 | 0.346 | 0.534 |
| DMPlug | 22.08 | 0.544 | 0.437 | - | - | - | 16.17 | 0.473 | 0.481 |
| **NA-NHMC (ours)** | **24.89** | **0.705** | 0.317 | 16.17 | 0.434 | 0.570 | **26.61** | **0.793** | **0.271** |

Table 15: Non-linear IPs on ImageNet ($256 \times 256$) with Gaussian Noise $\sigma_y = 0.20$. (**Bold**: best, underline: second best)

| | Nonlinear Deblurring | | | HDR Reconstruction | | |
|---|---|---|---|---|---|---|
| | PSNR ↑ | SSIM ↑ | LPIPS ↓ | PSNR ↑ | SSIM ↑ | LPIPS ↓ |
| DiffPIR | 21.52 | 0.492 | 0.526 | 19.94 | 0.556 | 0.418 |
| RED-diff | 12.47 | 0.071 | 0.759 | 16.46 | 0.267 | 0.593 |
| DPS | 16.11 | 0.340 | 0.551 | 21.92 | 0.519 | 0.448 |
| DAPS | 17.84 | 0.201 | 0.617 | 18.08 | 0.351 | 0.536 |
| SITCOM | 14.49 | 0.156 | 0.668 | 19.82 | 0.441 | 0.500 |
| DMPlug | 21.80 | 0.561 | 0.420 | 20.54 | 0.562 | 0.430 |
| **NA-NHMC (ours)** | **22.61** | **0.585** | **0.382** | **24.12** | **0.701** | **0.320** |

## A.10 ALTERNATIVE SAMPLING SCHEMES

Sampling in a space as high-dimensional as $(3 \times 256 \times 256)$ is a challenging task. Many standard sampling algorithms are not suitable in this setting. A key requirement for efficiency is the use of gradient information to accelerate convergence. The simplest such method is the Unadjusted Langevin Algorithm (ULA).

However, because ULA lacks a Metropolis–Hastings (MH) correction, its step size must be tuned carefully. Figure 9 illustrates this trade-off: large step sizes enable rapid exploration but cause significant discretization error as $\sigma_y$ approaches the target value, resulting in poor samples; conversely, small step sizes reduce error but lead to very slow exploration and long runtime.

Since different stages of the sampling chain require different effective step sizes, algorithms with a Metropolis–Hastings (MH) correction are more attractive, as the acceptance test provides a natural criterion for adapting step size. We therefore consider the Metropolis-Adjusted Langevin Algorithm (MALA), the No-U-Turn Sampler (NUTS), and Hamiltonian Monte Carlo (HMC). In practice, however, both MALA and NUTS tend to settle on excessively small step sizes in this high-dimensional setting, resulting in impractically long runtimes. By contrast, HMC accommodates larger step sizes and achieves a more favorable trade-off between accuracy and efficiency, making it the most suitable choice for our framework.

## A.11 ALTERNATIVE PRIOR MODEL: GAN-BASED INFERENCE

In place of the diffusion models, we experimented with StyleGAN2 (Karras et al., 2020) as the prior model for FFHQ $256 \times 256$ dataset. The quantitative results are presented in Table 17. The quality of image samples are significantly inferior to diffusion models across all tasks.

## A.12 ADDITIONAL QUALITATIVE RESULTS

In this section, we present additional qualitative results. Since we don't have access to an LDM for ImageNet ($256 \times 256$), ReSample cannot be applied to this dataset.

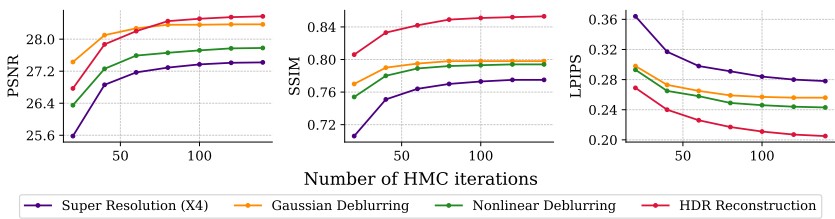

Figure 7: Performance of NA-NHMC across four tasks for FFHQ $(256 \times 256)$ as a function of the number of HMC iterations $K$. For all tasks, performance increases monotonically with more steps, but with diminishing improvements.

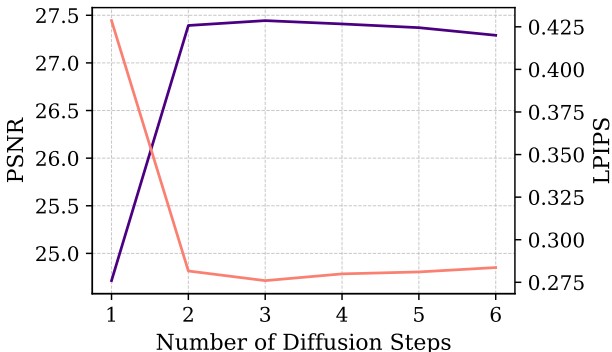

Figure 8: Performance of NA-NHMC on SR $(\times 4)$ task for FFHQ $(256 \times 256)$ as a function of the number of diffusion steps. The initial step is fixed at $T = 750$ for all cases to avoid numerical instability, and the remaining steps are evenly spaced in $[0, 750]$.

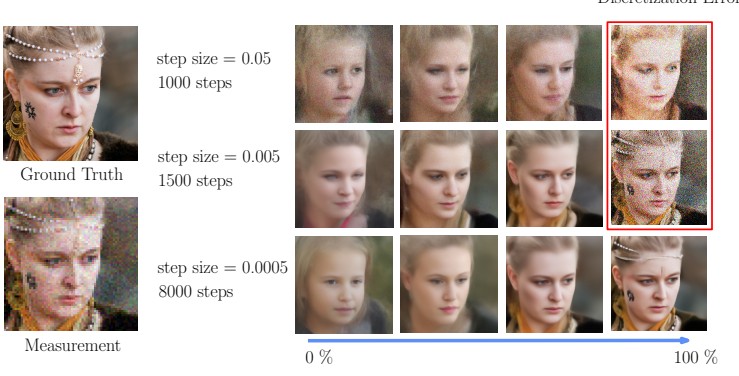

Figure 9: Unadjusted Langevin Algorithm (ULA) with different step sizes. Larger step sizes accelerate convergence but introduce greater discretization error, substantially degrading sample quality.

Table 16: Performance of NA-NHMC on SR $(\times 4)$ for FFHQ $(256 \times 256)$. Each schedule is defined by two parameters: (i) the first timestep (rows: $600, 750, 900$) and (ii) the final timestep (columns: $250, 375, 500$).

| Metrics | PSNR | | | SSIM | | | LPIPS | | |
|---|---|---|---|---|---|---|---|---|---|
| Schedule | 600 | 750 | 900 | 600 | 750 | 900 | 600 | 750 | 900 |
| 250 | 27.12 | 27.24 | 26.82 | 0.744 | 0.767 | 0.718 | **0.290** | 0.287 | 0.335 |
| 375 | 27.03 | **27.29** | 27.07 | 0.736 | **0.770** | 0.738 | 0.304 | 0.291 | 0.319 |
| 500 | 26.87 | 27.17 | 27.01 | 0.723 | 0.763 | 0.741 | 0.330 | 0.305 | 0.317 |

Table 17: GAN-Based Inference for Linear IPs on FFHQ (256 × 256) with Gaussian Noise $\sigma_y = 0.05$.

|  | Super Resolution (×4) | | | Random Inpainting (92%) | | |
| --- | --- | --- | --- | --- | --- | --- |
|  | PSNR ↑ | SSIM ↑ | LPIPS ↓ | PSNR ↑ | SSIM ↑ | LPIPS ↓ |
| NA-NHMC (GAN) | 18.27 | 0.454 | 0.513 | 17.80 | 0.440 | 0.528 |
| NA-NHMC (DDIM) | 27.29 | 0.770 | 0.291 | 26.72 | 0.785 | 0.268 |

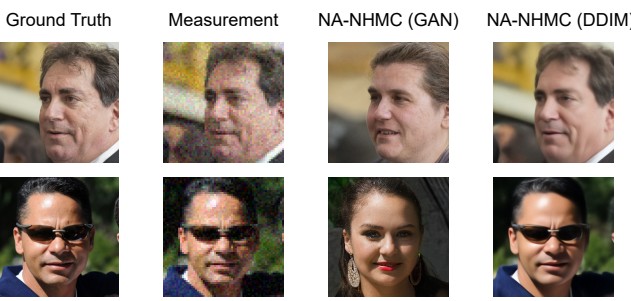

Figure 10: Comparison between GAN and Diffusion Model (DDIM) for SR(×4). FFHQ (256 × 256). $\sigma_y = 0.05$.

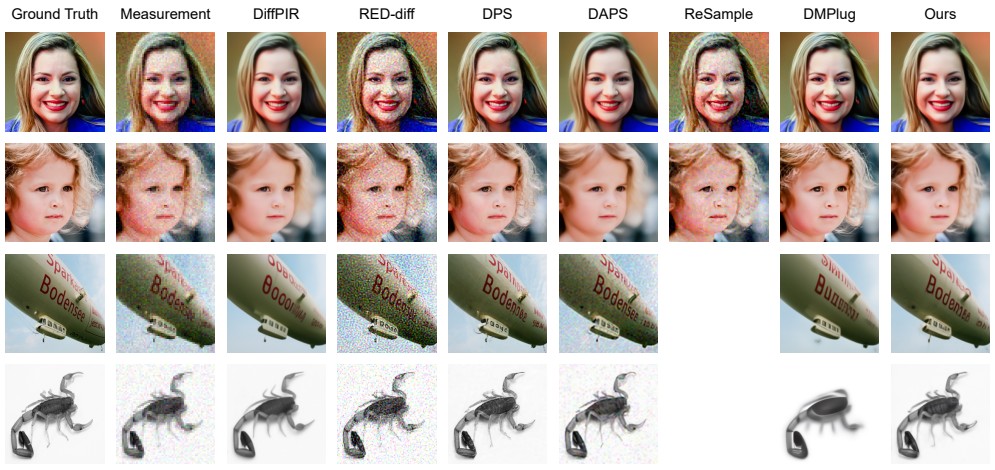

Figure 11: SR(×4). (Top) FFHQ (256 × 256). (Bottom) ImageNet (256 × 256). $\sigma_y = 0.05$

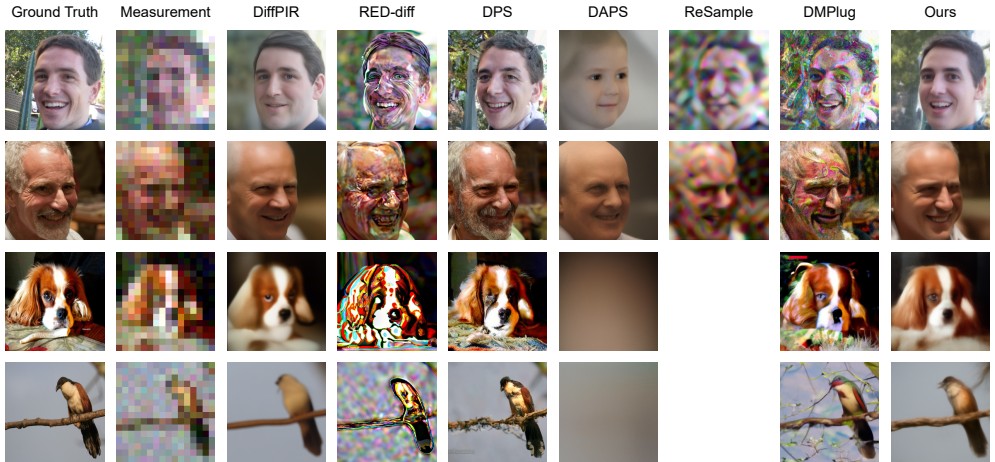

Figure 12: SR($\times$16). (Top) FFHQ ($256 \times 256$). (Bottom) ImageNet ($256 \times 256$). $\sigma_y = 0.05$

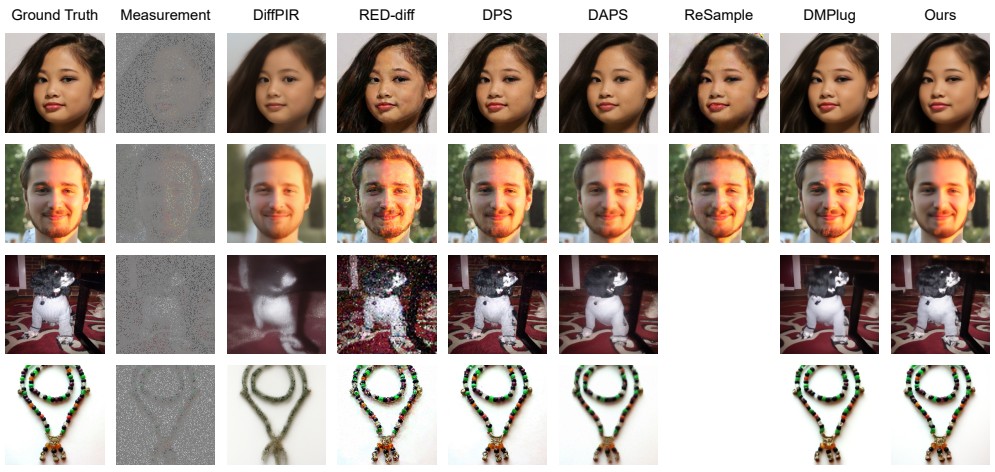

Figure 13: Random inpainting. (Top) FFHQ ($256 \times 256$). (Bottom) ImageNet ($256 \times 256$). $\sigma_y = 0.05$

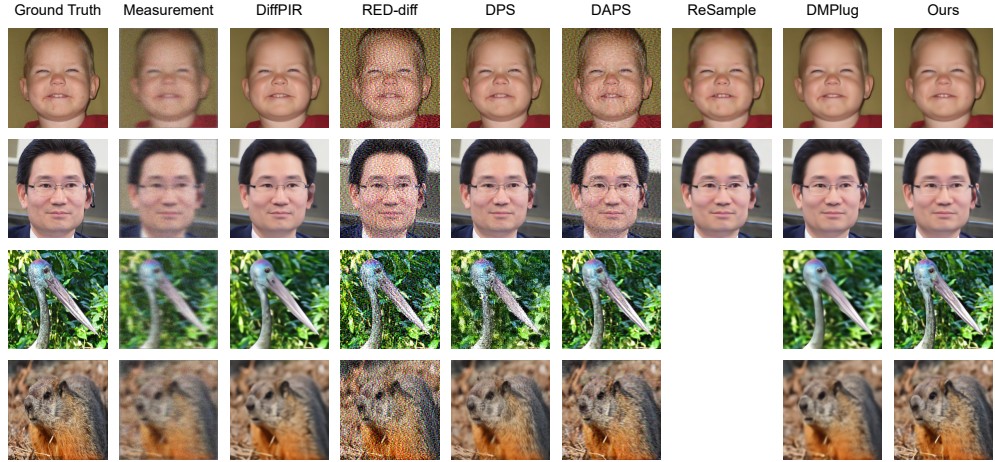

Figure 14: Gaussian deblurring. (Top) FFHQ ($256 \times 256$). (Bottom) ImageNet ($256 \times 256$). $\sigma_y = 0.05$

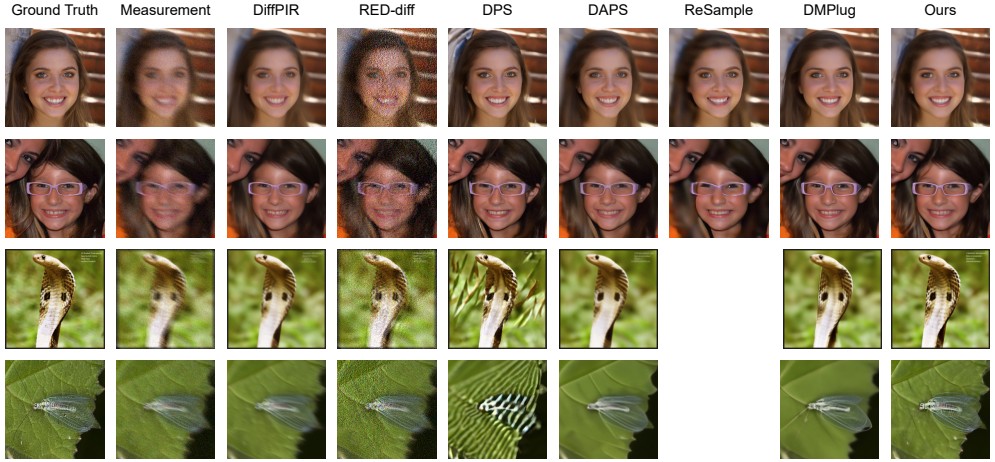

Figure 15: Nonlinear deblurring. (Top) FFHQ ($256 \times 256$). (Bottom) ImageNet ($256 \times 256$). $\sigma_y = 0.05$

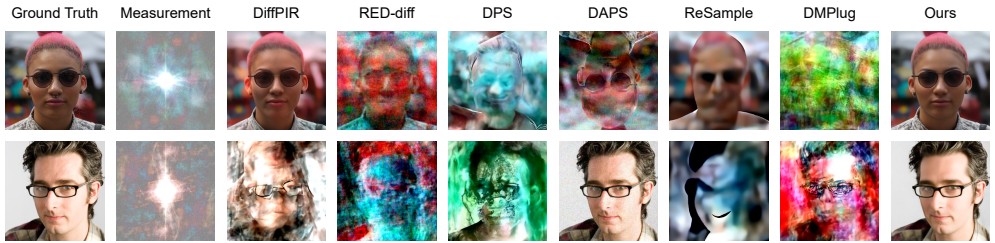

Figure 16: Phase retrieval. FFHQ ($256 \times 256$).

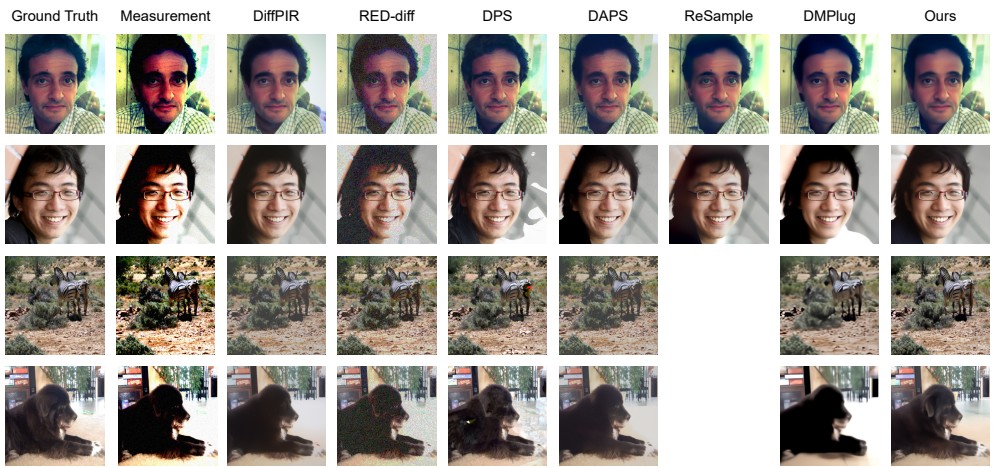

Figure 17: HDR reconstruction. (Top) FFHQ $(256 \times 256)$. (Bottom) ImageNet $(256 \times 256)$. $\sigma_y = 0.05$

