# OpenReview forum: "Noise-Adaptive Diffusion Sampling for Inverse Problems Without Task-Specific Tuning"
_ICLR.cc/2026/Conference — ICLR 2026 Poster_

### Official Review · Reviewer_eggp · 2025-10-15

**Soundness:** 2
**Presentation:** 2
**Contribution:** 2
**Rating:** 2
**Confidence:** 4

**Summary:**

The paper proposes Noise-space Hamiltonian Monte Carlo (N-HMC), which uses HMC to search for a good initial noise for solving inverse problems. The authors additionally propose a noise adaptive version of N-HMC, which adjusts the algorithm to work with unknown noise levels.

**Strengths:**

1. The performance seems to be good, outperforming several widely established baselines.

2. The method of using HMC for searching better noise initialization is new.

**Weaknesses:**

1. *What* N-HMC is solving is unclear. Is this doing posterior sampling? The mathematical statement should be precisely provided. Currently, the derivation starts with (7), which is ad-hoc. *Where* is the posterior scores used?

2. One of the motivations for this method is that the performance is inherently free form hyperparameter tuning, which does not seem to be the case. As the method is based on HMC, there are actually *more* hyperparameters that one can adjust, including how you would define the burn-in period. Reading the appendix, I am not convinced that the method requires less efforts for hparam tuning. It actually seems to require more effort, as opposed to methods such as DPS where one can just choose a step size.

3. In the experiments, two more metrics should be reported. PSNR/SSIM/LPIPS are all distortion metrics, and reporting them all does not give a more informed picture. 1) Report the FID values (perception metric) with more than 1k, 2) Report the computational cost. The computational cost is reported in the appendix, but it should be more accessible. 90 seconds is relatively slow, which is another drawback of the method.

4. The equality for $\nabla_{x_T} \log p(y|x_T)$ is, at best, an approximation. This holds across the entirety of the derivations.

5. The noise-adaptive part is confusing. $\sigma_y$ is undefined in the main text. It starts by stating that they model the noise variance with an inverse-gamma prior, which is arbitrary. How this leads to Alg. 3, is again, ambiguous. How is $m$ set?

6. Following 5, even for unknown noise levels, methods such as DPS are fine off with choosing a static step size (e.g. 1.0), which works well across all noise levels. If the authors were to truly argue that the noise adaptive part is important, then the experiments should be conducted on real-world degradations that are off the inverse crime setting.

**Questions:**

1. The authors assume that $\sigma_y$ follows an inverse-gamma prior, but later admits that they use an uniformative (i.e. uniform) prior. Any clarification on this?

2. Why is phase retrieval branded as a *multimodal IP*? All inverse problems are inherently multimodal, and this may confuse the readers.

---

> ### Author Response · Authors · 2025-11-27
>
> We thank all the reviewers for their thoughtful and constructive comments about our manuscript!
>
> - W1: is N-HMC doing posterior sampling?
>
> We confirm that N-HMC performs exact posterior sampling. Unlike standard methods that sample from the image posterior $p(x_0|y)$, we sample from the noise posterior $p(x_T|y)$. We treat the deterministic DDIM reverse process as a generator $\mathcal{D}: x_T \to x_0$. The target posterior distribution is defined as $p(x_T | y) \propto p(y | x_T) p(x_T) = p(y | \mathcal{D}(x_T)) \mathcal{N}(x_T; 0, I)$, where $p(x_T)$ is the standard Gaussian prior and $p(y | \mathcal{D}(x_T))$ is the likelihood of the generated image matching the measurements. Equation (7) is not ad-hoc; it is the score function (gradient of the log-density) of the posterior defined above. We compute it via Bayes' rule to drive the Hamiltonian dynamics. The posterior score derived in Eq. (7) is the "force" term used to update the momentum in the HMC algorithm. Specifically, it is used in Lines 8 and 11 of Algorithm 1: $p \leftarrow p - \frac{\delta}{2} \nabla_{x_T} \log p(x_T | y)$. We have revised Section 3.1 to explicitly define the posterior density $p(x_T|y)$ before deriving its gradient to improve clarity.
>
> - W2: "Hyperparameter-free" claim
>
> Thank you for this important question. Our claim is NOT "no hyperparameters" but rather "no TASK-SPECIFIC and NOISE-SPECIFIC tuning." The critical distinction is One-Time Setup vs. Per-Task Tuning: N-HMC (Ours): We use a single fixed set ($K=80, L=20, \delta_0=0.05, \gamma=0.95$) for apply to ALL 7 tasks, 2 datasets, multiple noise levels and type (low noise $\sigma_y=0.05$ and high noise $\sigma_y=0.20$, and Impulse and Speckle noise (Table 3)), as detailed in Appendix A.6. We added experiments in Appendix A.7 to show that performance is stable across a wide range of step sizes and leapfrog steps. In contrast, methods like DPS (Table 4 shows tuned learning rate $\zeta_i$ across tasks and dataset), DAPS (Task-specific $\eta_0$ values (Table 5): 1e-4, 1e-4, 1e-4, 1e-4, 5e-5, 5e-5, 2e-5) and DMPlug (Task-dependent early stopping with different window sizes and patience parameters) require tuning hyperparameters separately for each task to balance the likelihood and prior. Unlike traditional MCMC applications that require multiple independent samples for Monte Carlo estimation, image reconstruction prefers a high-quality sample. Instead of a traditional burn-in, we utilize a $\sigma_y$ annealing schedule as an automatic warm-up phase that efficiently guides $x_T$ from random initialization toward high-posterior regions. Figure 7 shows that reconstruction quality improves monotonically with K and starts to plateau around $K \approx 120$; using $K=80$ already lies in the stable regime and provides a good trade-off between quality and runtime.  We have clarified our claim in the revision to avoid misunderstanding.
>
> - W4: The  derivations for $\nabla_{x_T} \log p(y|x_T)$ is unclear.
>
> We respectfully clarify that under our framework, the computation of $\nabla_{x_T} \log p(y|x_T)$ is exact, not an approximation. This is a fundamental distinction between our method (**N-HMC**) and prior methods like DPS.
>
> **Why it is exact:** As defined in Section 3.1, we utilize DDIM as a deterministic mapping function $\hat{x}_0 = \mathcal{D}(x_T)$. Consequently, the likelihood $L(x_T)$ is explicitly:
> $$
> L(x_T) = -\frac{1}{2\sigma_y^2} \|y - \mathcal{A}(\mathcal{D}(x_T))\|^2
> $$
> The gradient of this term with respect to $x_T$ is computed using standard automatic differentiation (backpropagation) through the N-step ODE solver. This yields the true mathematical gradient of the defined objective function.
>
> **Comparison:**
> Methods like DPS introduce approximations (e.g., Tweedie’s formula or Taylor expansions) because they attempt to estimate the likelihood score $\nabla_{x_t} \log p(y|x_t)$ for intermediate noisy states. Our method bypasses this entirely by operating in the initial noise space, where the gradient chain is analytically complete.
>
> **The only "approximation":**
> The only approximation present is the modeling capacity of the pre-trained network $\mathcal{D}$ itself (i.e., how well it represents the true data distribution), which is a premise shared by all generative models. However, the inference process (the gradient computation and sampling) contains no mathematical approximations.
>
> We have updated Section 3.1 before Eq (7) and after Eq (9) to clarify that the gradient $\nabla_{x_T} \log p(y|x_T)$ is obtained via automatic differentiation through the deterministic generator, distinguishing our exact inference procedure from approximation-based guidance methods.
>
> **(continue)**

---

> > ### Author Response · Authors · 2025-11-27
> >
> > **(continue)**
> >
> > - W3: Missing FID scores and accessible computational cost comparison
> >
> > We thank the reviewer for suggesting additional metrics. We provide both FID results and detailed runtime analysis below.
> >
> > FID Results (computed on 1,000 FFHQ samples,  $\sigma_y=0.05$)
> > Task            | N-HMC | DPS   | DiffPIR | ReSample
> > ----------------|-------|-------|---------|----------
> > SR (×4)         | 38.8  | 21.2  | 46.0    | 77.9
> > Gaussian Deblur | 37.2  | 24.1  | 41.3    | 81.5
> > Nonlinear Deblur| 31.1  | 27.7  | 41.9    | 88.0
> > Phase Retrieval | 91.1  | 155.5 | 112.2   | 184.5
> > Inpainting      | 36.9  | 23.1  | 64.2    | 100.5
> > HDR             | 24.0  | 15.2  | 23.0    | 59.9
> >
> > DPS achieves a lower FID than NA-NHMC except for phase retrieval, whereas NA-NHMC consistently improves over ReSample and is generally comparable to DiffPIR. This reveals an important trade-off between reconstruction accuracy (PSNR/SSIM) and distribution-level perceptual quality (FID). The goal of inverse problems is accurate reconstruction of a specific ground truth image, not generating diverse samples from a distribution. We respectfully clarify that LPIPS is also a perceptual metric (learning-based), which can evaluate the single-image reconstruction, while FID is usually applied in unconditional generation to measure distribution-level similarity. The FID metric can be low even when individual reconstructions are inaccurate. For inverse problems where ground truth exists, PSNR/SSIM are the primary metrics, with FID providing complementary information.
> >
> > Runtime per image (FFHQ 256×256, NVIDIA H200):
> > Method      | Runtime | Algorithm Type        | Key Characteristics
> > ------------|---------|----------------------|---------------------
> > DPS         | ~23s    | Iterative guidance   | 1000 steps
> > DAPS        | ~56s    | Annealed MAP    | 250 steps + MCMC
> > DMPlug      | ~71s    | Optimization     | Requires early stopping
> > NA-NHMC     | ~90s    | Noise-space sampling   | 80 HMC iters × 2 steps
> > ReSample    | ~94s    | Image-space opt.     | 500 steps + latent optimization
> >
> >
> > NA-NHMC needs more computational budget,  since it performs rigorous posterior sampling (HMC), allowing for long-range exploration. It is easy to escape local minima where MAP methods like DAPS, DMPlug, and ReSample failed, especially in highly ill-posed tasks such as Phase Retrieval. In Table 1, we show that other fast sampling methods, such as DPS, don't perform well on nonlinear and high-noise inverse problems and require extensive time beforehand to tune task-specific hyperparameters (e.g., step sizes $\zeta_i$) for every new dataset, task, or noise level. Our method is robust to noise and task-specific tuning. Furthermore, all proposals are guaranteed to lie on the learned data manifold via unconditional DDIM, avoiding artifact issues shown in Figure 1. We have explicitly acknowledged this speed-accuracy trade-off as a limitation in the Conclusion and highlighted acceleration as future work.
> >
> > - W5\Q1: Clarify the noise-adaptive and inverse-gamma vs uninformative prior contradiction.
> >
> >
> > The choice of the Inverse-Gamma distribution is not arbitrary; it is the conjugate prior for the variance of a Gaussian likelihood. This is a standard choice in Bayesian inference that allows for analytically tractable marginalization of the unknown variance parameter. There is no contradiction. The uninformative prior we use (Jeffreys prior, $p(\sigma_y^2) \propto 1/\sigma_y^2$) is mathematically the limiting case of the Inverse-Gamma distribution ($\text{Inv-}\Gamma(\alpha, \beta)$) as the hyperparameters $\alpha \to 0$ and $\beta \to 0$. In the main text (and for Algorithm 3), we utilize this uninformative limit to ensure the method is fully parameter-free and requires no task-specific tuning. $m$ is the dimension of the measurement $y$. We have rewritten the relevant paragraph in Section 3.3 to clearly articulate this connection.
> > Algorithm 3 implements the Hamiltonian dynamics using the gradient of the marginalized posterior. By integrating out $\sigma_y^2$ using the uninformative prior, the log-likelihood becomes $\log p(y|x_T) = -\frac{m}{2} \log \left( \frac{1}{2} \|y - \mathcal{A}(\mathcal{D}(x_T))\|^2 \right) + C$ Taking the gradient with respect to $x_T$ yields the exact term used in Algorithm 3 $\nabla_{x_T} \log p(y|x_T) = -\frac{m}{2 \|y - \mathcal{A}(\mathcal{D}(x_T))\|^2} \nabla_{x_T} \|y - \mathcal{A}(\mathcal{D}(x_T))\|^2$. This term allows the sampler to automatically adapt the step size based on the measurement dimension ($m$) and the current residual, eliminating the need for a fixed $\sigma_y$.
> >
> > We have revised Section 3.3 to address these points.
> >
> > **(continue)**

---

> > > ### Author Response · Authors · 2025-11-27
> > >
> > > **(continue)**
> > >
> > > - W6: Clarify the importance of the noise adaptive part in solving the inverse problem. DPS are fine off with choosing a static step size (e.g., 1.0), which works well across all noise levels.
> > >
> > > We respectfully disagree with the premise that baseline methods like DPS are robust to hyperparameter choices across different settings. We provide evidence from our experiments to clarify why the noise-adaptive mechanism is critical. The reviewer suggests that a static step size (e.g., 1.0) works well for DPS across settings. However, our extensive grid search for DPS hyperparameters, reported in Table 4 (Appendix A.5), contradicts this. DPS is highly sensitive to the scale of the gradient, whereas NA-NHMC uses a single set of hyperparameters for all tasks and noise levels, effectively solving the tuning problem.
> > > For experiments on degradations "off the inverse crime setting." We have arguably already provided this in Section 4.3 and Table 3, where we tested on Impulse Noise and Speckle Noise. These noise types represent unknown, non-Gaussian statistics often found in real-world sensor errors. As shown in Table 3, DPS (tuned for Gaussian noise) degrades significantly under Impulse noise (e.g., SRx4 PSNR drops to 21.99 dB). In contrast, NA-NHMC adapts automatically and achieves 23.42 dB without any re-tuning. This demonstrates that our noise-adaptive formulation is not just a theoretical convenience but a practical necessity for handling unknown or non-standard degradation models where static baselines fail.
> > > The noise-adaptive term in Algorithm 3 scales the guidance gradient by the inverse of the current residual norm $\approx 1/\|y - \mathcal{A}(\hat{x})\|^2$. This acts as an automatic schedule: providing stronger guidance when the measurement consistency is low and weaker guidance (preventing overfitting) as the solution converges. This mechanism allows our method to transition seamlessly between high-noise ($\sigma=0.20$) and low-noise ($\sigma=0.01$) regimes, and even distinct noise types, which fixed-step methods cannot do.
> > > We have updated Section 4.1 to explicitly highlight the tuning sensitivity of DPS to reinforce this point.
> > >
> > > - Q2: Multimodal Definition.
> > >
> > > We appreciate the reviewer’s concerns. Regarding the terminology, we agree that, strictly speaking, inverse problems are multimodal. What we intended to emphasize is that phase retrieval is a particularly severely ill-posed nonlinear problem in which different modes (e.g., phase ambiguities, sign/shift aliases) can correspond to perceptually very different images despite similar measurements, making local-mode trapping especially problematic in practice. We revised the paper accordingly to avoid confusion.

---

> > > > ### Comment · Reviewer_eggp · 2025-11-28
> > > >
> > > > I thank the reviewers for clarifying many of the raised concerns. I still have further questions/concerns:
> > > >
> > > > 1. The authors' claim that the proposed method performs exact posterior sampling up to the approximation of the diffusion model is only partially right. The reason I am confused is because sampling from $p(x_0|x_T)$ is said to be equivalent to $\mathcal{D}(x_T)$, but with very few steps of discretization (e.g. 2 steps). The discretization error that arises from this process cannot be ignored. Is there any theoretical guarantee that this approximation error is indeed smaller than those methods that approximate the posterior score?
> > > >
> > > > 2. "For inverse problems where ground truth exists, PSNR/SSIM are the primary metrics, with FID providing complementary information.": I respectfully disagree. There is an inevitable trade-off between distortion and perception. When the goal is to minimize distortion, posterior sampling is *not* what one should do. Taking the posterior mean would suffice. This contradicts what the method is trying to do.
> > > >
> > > > While these concerns remain, since the other concerns were addressed, I raise my score to 4.

---

> > > > > ### Author Response · Authors · 2025-12-03
> > > > >
> > > > > Dear Reviewer eggp,
> > > > >
> > > > > Thank you for raising your score!
> > > > >
> > > > > - Q1
> > > > >
> > > > > We acknowledge that a 2-step process is a discretization of the continuous SDE. However, our method's strength lies in a specific design choice:
> > > > > 1. Clarification: You are correct that a 2-step DDIM is a discretization of the continuous diffusion process. However, in the N-HMC framework, we do not treat the 2-step process as an approximation of the 1000-step prior. Instead, we explicitly define the 2-step deterministic mapping $\mathcal{D}$ as our generative prior.
> > > > > **Our Method:** The generative model is fixed (albeit with limited capacity due to few steps). Given this model, our HMC sampling targets the exact posterior $p(x_T|y) \propto p(y|\mathcal{D}(x_T))p(x_T)$ using exact gradients via automatic differentiation. There is zero inference approximation error.
> > > > > **Baselines (e.g., DPS):** While they use a high-capacity (1000-step) model, they cannot compute the exact likelihood score $\nabla \log p(y|x_t)$. They must rely on approximations (e.g., Tweedie’s formula) at every single step. These errors accumulate, causing the trajectory to drift off the data manifold (Manifold Infeasibility).
> > > > >
> > > > > 2. Theoretical View: Modeling Bias vs. Inference Variance The comparison is effectively between "Exact Inference on a Simplified Model" (N-HMC) vs. "Approximate Inference on a Complex Model" (DPS). The "discretization error" you mentioned essentially manifests as a limit on generator capacity. Experiments demonstrate that, leveraging the powerful capabilities of pretrained diffusion models, our method and experiment settings are feasible.
> > > > >
> > > > > 3. Empirical Justification: This trade-off is validated by Figure 8 in our revised paper. Results show that performance plateaus at 2-3 steps. Increasing steps to reduce discretization error does not improve reconstruction quality, proving that the 2-step generator is sufficient for these inverse problems.
> > > > >
> > > > >
> > > > > - Q2
> > > > >
> > > > > Thank you for this insightful comment regarding the perception-distortion trade-off. We fully agree with the theoretical premise that, for a fixed posterior, minimizing distortion (MSE) is best achieved by point estimates like the posterior mean, whereas sampling introduces variance to improve perceptual quality.
> > > > > However, we believe NA-NHMC is the superior choice for these inverse problems for three specific reasons, which reconcile our high PSNR/SSIM scores with the sampling paradigm:
> > > > > 1. Robustness via Exploration (Avoiding Local Minima): As you rightly pointed out, point estimation (MAP/Optimization) should theoretically minimize distortion. However, in severely ill-posed or non-convex problems like Phase Retrieval, deterministic optimization often gets trapped in poor local modes due to the complex landscape. By performing posterior sampling with an annealing schedule for sigma_y, NA-NHMC encourages exploration early in the trajectory. This allows the solver to escape suboptimal modes and converge to the high-probability region of the true solution. This is why we observe significantly higher PSNR in Phase Retrieval (Table 1) compared to deterministic methods like DMPlug—not because we sacrifice perception, but because we successfully find the global mode where baselines fail.
> > > > >
> > > > > 2. Parameter-Free Balance: Standard MAP or guidance methods often require careful tuning of the weight between the likelihood and the prior to avoid collapsing onto noisy observations (overfitting) or drifting off-manifold. NA-NHMC avoids this issue naturally. The posterior score driving our HMC sampler directly encodes both terms derived from the probabilistic graphical model without introducing tunable weights. As illustrated in Figure 2, this prevents the sampler from collapsing onto noisy observations, naturally placing our results on a superior Pareto frontier where both fidelity (PSNR) and perception (LPIPS) are improved simultaneously.
> > > > >
> > > > > 3. Role of FID vs. LPIPS in Inverse Problems: We acknowledge FID is a standard metric for unconditional generation. However, in specific inverse problems (like deblurring specific faces), FID can sometimes be misleading because it measures the distance between distributions, not the perceptual alignment of specific sample pairs. A method could hallucinate realistic but wrong details (low FID, very low PSNR), which is undesirable for reconstruction tasks. We believe reporting PSNR (fidelity), SSIM (structure), and LPIPS (perception) provides a holistic view. The fact that we achieve SOTA results in both PSNR and LPIPS suggests our sampler accurately locates the high-probability mode of the posterior without collapsing to a blurry mean.

---

### Official Review · Reviewer_kd9g · 2025-10-29

**Soundness:** 3
**Presentation:** 3
**Contribution:** 2
**Rating:** 6
**Confidence:** 4

**Summary:**

To avoid local minima or noise overfitting problem in inverse problem solving with diffusion model, the paper proposed to search a good initial noise by Hamiltonian Monte Carlo (HMC), which leads to the following sampling through an ODE staying on the data manifold. For this, the proposed method repeat updating initial noise with sampling only with 2 denoising steps. The paper also introduces noise adaptive sampling which provides robustness on various measurement noises.

**Strengths:**

- The paper presents motivation and methods clearly.
- The paper considers various measurement noise including impulse and speckle noises, which increases the effectiveness of the proposed method in real world.
- Extensive experiments support the effectiveness of the proposed method and gives sufficient analysis on its behavior.

**Weaknesses:**

- The major difference from DAPS is twofolds: the paper uses HMC instead of Langevin dynamics, and the search space is changed from image to noise space. However, both changes seems to introduce additional computational cost, which results in slower sampling.
- Missing related work [1] that update the initial noise with data fidelity gradient after sampling.
- The performance reported in Table 1 has a huge gap from the original paper. For example, DAPS for Phase Retrieval originally achieves 30.63dB of PSNR with the same setting, but it is 18.52dB in this paper.


References

[1] Diffusion Image Prior, ICCV 2025

**Questions:**

- Could authors explain the reason of large gap of performance between the original baseline paper and this paper?
- Could authors provide runtime comparison by setting the same number of function evaluation? Or Could authors provide performance comparison by setting the same computational budget?
- Is there any reason for empty boxes for ReSample in Figure 10 - 16?
- What if we use the GAN or consistency model instead of diffusion model? The reviewer cannot find a strong reason to use the diffusion model from the algorithm 1.

---

> ### Author Response · Authors · 2025-11-27
>
> We thank all the reviewers for their thoughtful and constructive comments about our manuscript! Below, we address your concerns regarding baselines, computational cost, and model choices.
>
> - W2: Missing related work Diffusion Image Prior that update the initial noise with data fidelity gradient after sampling.
>
> Thank you for pointing out this recent work. We were aware of Diffusion Image Prior (DIIP, Yang et al., ICCV 2025), and we apologize for not including it in our discussion. We cited and discussed this paper in sec 2.1.
>
> - Q1/W3: Reported performance for DAPS is much lower than in the original paper. Explanation for performance gap of DAPS baseline
>
> Thank you for catching this and for prompting us to clarify the evaluation protocol. The main reason for the discrepancy is that the original DAPS paper evaluates phase retrieval using a best-of-4 strategy: for each measurement, they run the sampler four times with an oversampling ratio of 2.0 and report the best PSNR among the four runs (please see experiment settings in DAPS and the accompanying code). In contrast, all results in our Table 1 are obtained from a single run per measurement for every method, in order to keep the evaluation protocol consistent across baselines and to reflect the success rate of a single posterior sample. Under this single-run setting, our DAPS reimplementation obtains 18.52 dB, which we reported in the paper. We will also release our code and configuration files to ensure that all baselines, including DAPS, can be reproduced under the same protocol.
>
> - W1/Q2: Runtime / computational budget comparison. Performance comparison by setting the same computational budget
>
> We have conducted a detailed runtime analysis on an NVIDIA H200 GPU to address the concern that these choices lead to inefficient sampling. We measured the average runtime per image ($256\times256$): 1. DPS: 23s 2. DAPS: 56s 3. NA-NHMC (Ours): 90s (for full convergence at 80 iterations). While our total runtime for full convergence (90s) is higher than DAPS (56s), our method is significantly more efficient in achieving high-quality results in some tasks. For example, the Super Resolution ($\times 4$) task on FFHQ in Figure 7, within 23 seconds (approx. 25 iterations), our method effectively plateaus, achieving a PSNR (26.8 dB) that matches DPS (26.84 dB0) and significantly outperforms DAPS (24.58 dB). It is correct that noise-space sampling adds computational cost, as computing the exact gradient $\nabla_{x_T} \log p(y|\mathcal{D}(x_T))$ requires backpropagation through the decoder. However, we believe this cost represents a necessary "robustness premium" that justifies the trade-off. Unlike faster but heuristic methods (e.g., DPS), our approach provides Noise Adaptivity for blind inverse problems, Zero Tuning stability across tasks, and Manifold Guarantees that prevent artifacts. Empirically, the exact gradient signal is highly efficient: NA-NHMC converges to superior solutions in fewer wall-clock seconds than heuristic approximations, notably outperforming baselines like DAPS in half the runtime.
>
> - Q3: Missing ReSample results in qualitative figures
>
> Yes—this is due to a limitation of the available pretrained models. In the original paper, line 1082, we mentioned that “Since we don’t have access to an LDM for ImageNet (256 × 256), ReSample cannot be applied to this dataset.”   ReSample is implemented on top of latent diffusion models and, to the best of our knowledge, a pretrained 256×256 ImageNet LDM is not publicly available. As a result, we can only run ReSample on FFHQ, but not on ImageNet.
>
> - Q4: Why diffusion models for the prior model?
>
> Conceptually, Algorithm 1 only assumes access to a differentiable generative mapping from a latent variable to images.
> In this work, we chose diffusion models as priors for two reasons:
> 1. Availability and comparability. High-quality 256×256 pretrained models for FFHQ and ImageNet are widely available for diffusion models (Chung et al. 2023; Dhariwal & Nichol 2021), and all of our main baselines (DPS, DAPS, DMPlug, ReSample) are defined in the diffusion framework. Using the same diffusion priors ensures a fair comparison across methods.
>
> 2. Empirical performance on inverse problems. Recent work has shown that diffusion models provide particularly strong priors for both linear and nonlinear inverse problems, especially under high noise and complex degradations. Our results further support this observation (Tables 1–3). In appendix A.11, we show that the quantitative results when using GAN as the prior model are significantly inferior to those of diffusion models.
> We clarified in the revised manuscript that N‑HMC is prior-agnostic and can, in principle, be combined with other generative architectures such as GANs or consistency models.

---

### Official Review · Reviewer_4fZg · 2025-11-01

**Soundness:** 2
**Presentation:** 2
**Contribution:** 2
**Rating:** 2
**Confidence:** 4

**Summary:**

This paper proposes a noise-adaptive method for solving inverse problems using diffusion models as priors. The main goal of the paper is to develop methods that adapt to the manifold structure of data, hence obtaining better performance for inverse problems under noisy scenarios.

**Strengths:**

This paper addresses an important problem, that is the noise sensitivity and intractability of general methods used for solving inverse problems via diffusion priors.

**Weaknesses:**

The paper has a few issues that needs to be addressed comprehensively (see questions for details):

- the mathematical justification is thin and derivations are unclear
- experimental results are missing some baselines proposed to address the same issues, notably noise sensitivity of inverse problem solvers.
- the method is inherently expensive as DDIM mapping should be autodiffed - the paper avoids this by using a few steps, which is not very principled.

**Questions:**

- Why is the method called noise-space sampling? This is a bit confusing.

- In general, there is a lot of mentions of "manifold" but I found this non-rigorous. There's really no geometric insight in any of these comments. For example, the authors mention "manifold feasibility problem" as the main motivation of their work, but this is not defined or explained. Is there any theoretical result regarding this?

- Please clarify the equation in line 211. How does first equality work? Since $\mathcal{D}(x_T) \approx x_0$, I don't understand why the first equality in line 211 would work.

-  Please provide a remark after Proposition 1 that explains and clarifies the result.

- The paper misses a reference for comparison, which also addresses the noise-robustness issue of standard solvers by adopting a second-order view:

> *Boys, B., Girolami, M., Pidstrigach, J., Reich, S., Mosca, A., & Akyildiz, O. D. Tweedie Moment Projected Diffusions for Inverse Problems. Transactions on Machine Learning Research, 2024.*

Please add this benchmark to your comparisons in your experiments.

- To see the introduced bias of the method in a simple setting, the paper would benefit from a simple experiment, see Figure 1 of the paper cited above. Please consider adding this.

style comment: I do not think using bold text in such frequency is appropriate -- in fact, I think standard academic writing only allows italics for emphasizing - no bolds please.

---

> ### Author Response · Authors · 2025-11-27
>
> We thank all the reviewers for their thoughtful and constructive comments about our manuscript!
>
> - Q1: why is the method call noise-space sampling?
>
> We apologize for any confusion caused by this terminology and thank the reviewer for pointing it out. We use the term "noise-space sampling" to highlight that our HMC algorithm operates exclusively on the initial variable $x_T \sim \mathcal{N}(0, I)$, which serves as the input latent variable for the diffusion model. Unlike iterative guidance methods (e.g., DPS) that modify intermediate noisy images $x_t$, our method treats the entire reverse diffusion process as a deterministic mapping $\mathcal{D}: x_T \mapsto \hat{x}_0$ (via DDIM). By sampling directly in the space of $x_T$, which follows a simple Gaussian distribution, we ensure that every generated proposal naturally adheres to the pre-trained data distribution, as any $x_T$ mapped through $\mathcal{D}$ lies on the learned manifold by definition.
> We clarified this terminology in the revised Sec 3.1 to avoid confusion.
>
> - Q2: "Manifold Feasibility Problem" Definition and Explanation
>
> We thank the reviewer for pointing out the need for a rigorous definition. We have revised Section 3.1 to define and explain this concept formally.
>
> Formal Definition: Following Alkhouri et al. (SITCOM, ICML 2025), we define Manifold Feasibility as the requirement that intermediate states $\mathbf{x}_t$ during sampling must remain within the high-probability generative manifold $\mathcal{M}_t$ of the marginal distribution $p_t(\mathbf{x}_t)$.
>
> Geometric Insight: Standard methods (e.g., DPS) violate this because they update $x_t$ using a likelihood gradient $\nabla_{x_t} \log p(y|x_t)$. Geometrically, this gradient vector often points in directions orthogonal to the thin data manifold in high-dimensional space. By taking a step in this direction, the state $x_t$ is pushed "off-manifold" into a low-probability region. This forces the pre-trained denoiser $\epsilon_\theta(x_t, t)$ to process out-of-distribution inputs, leading to the accumulated artifacts observed in Figure 1(a).
> N-HMC guarantees the manifold feasibility. We limit the search space exclusively to the initial noise $x_T \sim \mathcal{N}(0, I)$. Since we use the unconditional DDIM mapping $\mathcal{D}: x_T \to x_0$ as a deterministic generator, any $x_T$ we sample maps to a valid point on the learned manifold $\mathcal{M}_\theta$. This ensures our reconstructed $x_0$ is always "on-manifold," avoiding the artifacts seen in guidance-based methods as shown in Figure 3.
>
> [1] Sitcom: Step-wise triple-consistent diffusion sampling for inverse problems. ICML 2025.
>
> - W1/Q3: derivations are unclear. Line 211 Derivation.
>
> We appreciate the opportunity to clarify this key formulation. We understand that identifying $\mathcal{D}(x_T)$ with $x_0$ may appear to be an approximation if viewed through the lens of stochastic diffusion paths. However, in our proposed N-HMC framework, this equality is mathematically exact by definition.
>
> Here is the rigorous justification:
> 1. **Deterministic Generator Definition:** We explicitly define our generative model using unconditional DDIM as a deterministic function $\mathcal{D}(\cdot)$. In this model, the initial noise $x_T$ is the only latent variable.
> 2. **Exact Likelihood:** The mapping from the latent code $x_T$ to the measurement $y$ follows the deterministic chain: first $x_T$ is mapped to \hat{$x_0$} via $\mathcal{D}$, then \hat{$x_0$} is mapped to $y$ via the forward operator $\mathcal{A}$ and noise $\eta$. The likelihood is well-defined via this deterministic transformation as $p(y|x_T) := p(y|\mathcal{D}(x_T))$. Since $\mathcal{D}$ is deterministic, knowing $x_T$ completely determines $x_0 = \mathcal{D}(x_T)$, and thus determines the distribution of $y$ via the measurement model $y = \mathcal{A}(x_0) + \eta$. Bayes' rule applies exactly. The gradient $\nabla_{x_T} \log p(y|x_T)$ is computed exactly via automatic differentiation through the deterministic DDIM steps—no approximation is introduced.
> 3. **Regarding $D(x_T) \approx x_0$:** This notation means that $D$ is the learned approximation to the true data distribution. However, given the trained diffusion model $D$, the mapping from $x_T$ to $D(x_T)$ is exact and deterministic. The approximation error is in how well the learned model $D$ represents the true data manifold, not in our inference procedure. However, the inference process (the gradient computation and sampling) contains no mathematical approximations.
>
> We have revised Section 3.1 to explicitly define this probabilistic graphical model ($x_T \to \hat{x}_0 \to y$) and emphasize that $\mathcal{D}$ is treated as a deterministic generator, ensuring the derivation is mathematically rigorous.

---

> > ### Author Response · Authors · 2025-11-27
> >
> > (continue)
> > - W3: A few steps using DDIM
> >
> > We do not treat the 2-step process as a "poor approximation" of the 1000-step diffusion. Instead, we formally define our generative prior $\mathcal{D}$ as a deterministic mapping composed of 2 DDIM steps. Our method performs exact Bayesian posterior sampling with respect to this specific generator. As long as this generator outputs high-quality samples on the manifold (which our results confirm), the inference framework remains mathematically rigorous.
> > We agree that backpropagating through the ODE is expensive. As shown in our ablation study (Figure 8, Appendix A.8), reconstruction quality (PSNR/LPIPS) plateaus after 2-3 steps. Increasing diffusion steps does not yield significant gains but linearly increases computational cost. 2 steps is the optimal operating point, balancing generation quality and sampling efficiency.
> >
> > - Q4: Remark after Proposition 1 that explains and clarifies the result
> >
> > We have added a formal Remark immediately following Proposition 1 in Section 3.2 of the revised paper. This remark provides a key interpretation of the derived equation: the expected residual decomposes into two distinct sources—(1) the measurement noise variance, and (2) the intrinsic uncertainty of the diffusion prior projected onto the measurement space. This decomposition theoretically justifies the robustness of N-HMC. Unlike optimization-based methods that may drive the residual to zero (leading to noise overfitting), our posterior sampling approach inherently maintains a residual magnitude consistent with the true noise level $\sigma_y$.
> >
> >
> >
> > - W2/Q5/Q6: Missing Baseline on noise sensitivity of inverse problem solvers-Tweedie Moment Projected Diffusions addresses the noise-robustness issue by adopting a second-order
> >
> > We sincerely thank the reviewer for pointing out this important reference. TMPD is indeed highly relevant as it addresses noise robustness through a second-order perspective. We have updated our manuscript (Section 1,2) to cite and discuss TMPD. We made extensive efforts to reproduce TMPD results using the official implementation: We made a significant effort to reproduce the official tmpdjax implementation to provide the requested quantitative comparison and bias visualization (Figure 1 of Boys et al.). Unfortunately, we were unable to run the evaluation due to severe library incompatibilities with our hardware environment, despite testing multiple combinations of Python (3.8, 3.9, 3.10) / JAX (0.2.x, 0.3.x, 0.4.x) / diffusionjax (0.1.x) versions. The recommended versions (Python 3.8 + JAX 0.2.x) are incompatible with our CUDA 12.7 / H200 GPU infrastructure. Compatible versions (Python 3.9 + JAX 0.3.x) trigger the std method error shown: AttributeError: 'VE' object has no attribute 'std' within the score_fn utility. We are actively working with the authors' codebase and will continue attempts to resolve these compatibility issues. If successfully implemented, we will include TMPD comparisons in the final version.
> >
> >
> > - Q7: style comment: italics for emphasizing
> >
> > We apologize for the excessive use of bold text. We have revised the manuscript to use italics for emphasis, adhering to standard academic style.

---

### Official Review · Reviewer_iaDC · 2025-11-01

**Soundness:** 3
**Presentation:** 3
**Contribution:** 3
**Rating:** 8
**Confidence:** 3

**Summary:**

This paper presents Noise-space Hamiltonian Monte Carlo (N-HMC) sampler, using a pretrained diffusion prior to solve general inverse problems. N-HMC directly samples from $p(x_T|y)$ with a few-step diffusion rollout estimating $\hat x_0^*$. While reminiscent of DMPlug, it formulates recovery as sampling rather than optimization, naturally accommodating measurement noise. The authors establish theoretical guarantees for N-HMC, showing its robustness to measurement noise under mild assumptions. Based upon N-HMC, NA-NHMC is proposed to adapt to the possibly unknown measurement noise level without hyperparameter tuning. Experiments on natural images show consistent gains over prior methods, with especially strong performance in noisy inverse-problem settings.

**Strengths:**

- This paper is overall well-written. It categorizes and clearly explains the strengths and weaknesses of existing methods, especially highlighting how and why existing methods are sensitive to measurement noise and rely on extensive hyperparameter tuning.
- The proposed N-HMC sampler is well justified by Proposition 1 that indicates its robustness to measurement noises.
- NA-NHMC extends N-HMC to a blind inverse problem setting where the noise level is unknown. NA-NHMC coincides with N-HMC with known noise level under inverse-gamma prior assumption of the noise level, which is demonstrated both in theory and in practice.
- Experimental results show clear advantage of NA-NHMC over existing diffusion posterior samplers on image restoration tasks, especially with varying measurement noise levels.

**Weaknesses:**

- I found no major weaknesses in this paper, but I believe some justifications are needed. See questions below.

**Questions:**

- The proposed N-HMC performs $p(x_T|y)$ sampling in the noisy space with the help of a few-step sampler that estimates $\hat x_0$. Similar sampling strategy is discussed in [1], which also samples in the noisy space but follows an noise annealing scheme as ReSample and DAPS. Can the authors comment on the differences between these methods? In particular, what are the pros and cons of sampling $p(x_T|y)$ vs. sampling $p(x_t|y)$ with an annealing noise schedule? Also, it seems an empirical comparison against SITCOM is necessary as it reported better results than DMPlug and DAPS in the considered experimental setups.
- What is the reason of choosing inverse-gamma prior for $\sigma_y$? Is it a specific trick to derive proposition 2?

[1] Alkhouri et al. "SITCOM: Step-wise Triple-Consistent Diffusion Sampling for Inverse Problems", ICML 2025.

---

> ### Author Response · Authors · 2025-11-27
>
> We sincerely thank the reviewer for the thoughtful and constructive feedback and recognition of the strengths of our paper.
> - Q1: Differences between N-HMC and existing noise-space sampling methods. Pros/cons of sampling $p(x_T |y)$ vs. $p(x_t|y)$ with annealing. Empirical comparison.
>
> We thank the reviewer for the positive assessment and for bringing SITCOM [1] to our attention. We have incorporated SITCOM into our baselines and updated the manuscript accordingly.
> Differences between N-HMC and SITCOM: 1. SITCOM performs step-wise MAP optimization on intermediate states $x_t$. It requires early stopping and noise-specific thresholds to prevent overfitting. N-HMC employs Hamiltonian Monte Carlo (MCMC posterior sampling) on $x_T$, which leverages momentum to enable efficient exploration of distant regions in the high-dimensional posterior in initial iterations. Furthermore, the NA-NHMC extension of our method allows for the sampler to estimate the measurement noise level based on the observed residual $y-A(x_0)$, eliminating the need for any prior knowledge of the measurement noise. 2. Manifold Consistency: Since N-HMC only updates $x_T$ and maps it through the deterministic $\mathcal{D}$, every proposal is guaranteed to lie on the valid data manifold defined by the pre-trained model. SITCOM modifies the trajectory at intermediate steps $t$. If the optimization pushes $v_t$ slightly off the manifold to satisfy data consistency, this deviation can persist, leading to "manifold infeasibility". 3. Annealing: SITCOM follows the standard diffusion noise schedule. N-HMC uses temperature annealing on the likelihood term $\sigma_y$ to flatten the posterior landscape initially, promoting global exploration before converging to the target distribution.
> Pros and Cons: SITCOM excels in linear IPs with known noise but requires task-specific tuning and risks noise overfitting in blind settings (Sec. 4.3). In contrast, NA-NHMC simultaneously samples the posterior and estimates noise levels, ensuring robustness to unknown or non-Gaussian noise. While computationally heavier, its HMC-based exploration guarantees strict manifold adherence and avoids the local optima often faced by optimization methods.
> Empirical Comparison (Updated in Tables 1, 2, 10-15): We compared NA-NHMC with SITCOM using their official implementation and recommended hyperparameters. SITCOM performs comparably to NA-NHMC in low-noise linear tasks. NA-NHMC significantly outperforms it in challenging scenarios. Specifically, NA-NHMC avoids mode collapse in multimodal phase retrieval (19.30 dB vs. 11.89 dB) and remains robust against overfitting in high-noise nonlinear deblurring ($\sigma_y=0.20$: 24.89 dB vs. 16.26 dB) without parameter tuning.
>
> - Q2: Justification for Inverse-Gamma Prior.
>
> The Inverse-Gamma prior is not a specific trick for Proposition 2, but the standard conjugate prior for Gaussian variance in Bayesian inference.
> Analytical Marginalization: Conjugacy allows us to analytically marginalize out $\sigma_y^2$, yielding a closed-form likelihood (Eq. 11) that effectively acts as a Student-t distribution (heavier tails). This avoids treating $\sigma_y$ as a hyperparameter to be tuned.
> Non-informative Limit: In high dimensions ($m \approx 2 \times 10^5$), the data term dominates the prior parameters ($\alpha, \beta$). This effectively behaves as a Jeffreys prior ($p(\sigma^2) \propto 1/\sigma^2$), ensuring the inference is data-driven rather than biased by the prior.
> Theoretical Alignment: Proposition 2 confirms that in this non-informative limit, the update rule of NA-NHMC mathematically converges to the exact N-HMC update rule derived when $\sigma_y$ is known.
> Empirical Proof: Figure 2 shows that our method accurately estimates the true noise level (e.g., recovering $\sigma_y=0.05$ from data) solely through this formulation, confirming the validity of this choice.
> We have clarified these points in Section 3.3 and Appendix A.1.

---

### Author Response · Authors · 2025-12-03
**General Response and Summary**

We thank reviewers and AC for the valuable feedback and time. We summarize the key recognitions and updates during the original review and rebuttal process below

---
***1. Strong motivation***
All reviewers acknowledge the significance of identifying three key issues in existing diffusion-based methods: manifold infeasibility, noise overfitting, and local mode collapse. The proposed noise-space sampling approach is recognized as a principled solution.

***2. Theoretical framework***
Reviewers recognize the well-justified methodology (iaDC), including Proposition 1's robustness guarantee and Proposition 2's theoretical alignment between NA-NHMC and N-HMC with known noise levels.

***3. Comprehensive experiments***
Reviewers acknowledge extensive experiments across 7 inverse problems (4 linear + 3 nonlinear) on two datasets, with consistent gains especially in challenging scenarios: phase retrieval, high-noise settings, and non-Gaussian noise types.

***4. No task-specific tuning***
Reviewers note that NA-NHMC uses a single hyperparameter configuration across all tasks, datasets, and noise levels—a significant practical advantage over baselines requiring per-task calibration.

---

**1. Mathematical rigor clarified. (Section 3.1 revised)**
Addressing 4fZg (W1, Q3), eggp (W1, W4)

***Exact gradient, not approximation:*** Clarified that $\nabla_{x_T}\log p(y|x_T)$ is computed exactly via automatic differentiation through deterministic DDIM—no Tweedie approximation involved.

***Formal definition added:*** Added Definition 3.1 for manifold feasibility with geometric explanation.
Inverse-gamma justification: Clarified Jeffreys prior as limiting case of inverse-gamma; no contradiction with uninformative prior claim.

**2. New baseline added.** (Tables 1, 2, 10-15) Addressing iaDC (Q1)

Added SITCOM (ICML 2025) to all experimental tables using official implementation. NA-NHMC significantly outperforms SITCOM in challenging scenarios: phase retrieval (19.30 dB vs. 11.89 dB), high-noise nonlinear deblurring (24.89 dB vs. 16.26 dB).

**3.Hyperparameter-free claim scoped.** (Section 3.3, Appendix A.6-A.7) Addressing eggp (W2), kd9g (W1)

1. Clarified claim as "no *task-specific* tuning": single configuration for all 7 tasks, 2 datasets, multiple noise types.

2. Added hyperparameter sensitivity analysis (Tables 7-9) showing stable performance across wide parameter ranges.

3. Contrasted with DPS/DAPS requiring per-task tuning (Tables 4-5).

**4. FID and runtime analysis reported** Addressing eggp (W3), kd9g (W1, Q2)

Clarified and explained metrics FID, PSNR/SSIM, and LPIPS in our experiments.

Runtime comparison: DPS ~23s, DAPS ~56s, NA-NHMC ~90s. Extra cost justified as "robustness premium", avoiding tuning and failure modes.

**5. Design choice clarifications.** Addressing eggp (W5, Q1, Q2), 4fZg (Q1, Q2), kd9g (Q4)

1. 2-step DDIM validity: Deliberately defined as generative prior; ablation (Figure 8) confirms 2-3 steps are sufficient.

2. Posterior sampling confirmed: N-HMC samples from the exact posterior

3. Prior-agnostic framework: N-HMC works with GANs (Appendix A.11 shows inferior results) or other generators.

**6. Baseline discrepancy explained.** Addressing kd9g (W3, Q1)

DAPS original paper uses best-of-4 evaluation; our tables report single-run results for fair comparison.
DIP (ICCV 2025) cited and discussed in revised Section 2.1.

**7. Terminology revised.** Addressing 4fZg (Q7), eggp (Q2)

Replaced bold emphasis with italics per academic style.
Revised "multimodal" to "severely ill-posed" for phase retrieval to avoid confusion.

---

### Meta-Review · Area_Chair_b5KL · 2026-01-06

**Summary:**

The paper proposes Noise-space Hamiltonian Monte Carlo (N-HMC) for diffusion-based inverse problems. The key idea is to use HMC to search for a good initial noise, rather than optimizing directly in image space. This is a clean and interesting perspective. The authors also present a noise-adaptive variant for unknown noise levels. It updates the initial noise repeatedly while using only two DDIM denoising steps, which keeps the method practical.

Reviewer eggp raises a fair point about the “exact posterior sampling” claim. The authors clarify that their argument relies on an assumption. In particular, they assume two-step DDIM can reliably reach the correct solution. The optimization is then built on top of that. This clarification helps, but the paper should state more clearly when two-step DDIM is expressive enough. This may depend on the diffusion model and the task.

Overall, I see this as a promising contribution with a reasonable evaluation. The weaknesses are mostly about positioning and clarity, rather than the core idea. I lean weak positive and recommend acceptance.

**Reviewer Concerns:**

.

**Reviewer Scores:**

.

---

### Decision · Program_Chairs · 2026-01-26

Accept (Poster)